# Genome mining unearths a hybrid nonribosomal peptide synthetase-like-pteridine synthase biosynthetic gene cluster

Hyun Bong Park[1,2†], Corey E Perez[1,2†], Karl W Barber[3,4], Jesse Rinehart[3,4], Jason M Crawford[1,2,5*]

[1]Department of Chemistry, Yale University, New Haven, United States; [2]Chemical Biology Institute, Yale University, West Haven, United States; [3]Department of Cellular and Molecular Physiology, Yale School of Medicine, New Haven, United States; [4]Systems Biology Institute, Yale University, West Haven, United States; [5]Department of Microbial Pathogenesis, Yale School of Medicine, New Haven, United States

**Abstract** Nonribosomal peptides represent a large class of metabolites with pharmaceutical relevance. Pteridines, such as pterins, folates, and flavins, are heterocyclic metabolites that often serve as redox-active cofactors. The biosynthetic machineries for construction of these distinct classes of small molecules operate independently in the cell. Here, we discovered an unprecedented nonribosomal peptide synthetase-like-pteridine synthase hybrid biosynthetic gene cluster in *Photorhabdus luminescens* using genome synteny analysis. *P. luminescens* is a Gammaproteobacterium that undergoes phenotypic variation and can have both pathogenic and mutualistic roles. Through extensive gene deletion, pathway-targeted molecular networking, quantitative proteomic analysis, and NMR, we show that the genetic locus affects the regulation of quorum sensing and secondary metabolic enzymes and encodes new pteridine metabolites functionalized with *cis*-amide acyl-side chains, termed pepteridine A (**1**) and B (**2**). The pepteridines are produced in the pathogenic phenotypic variant and represent the first reported metabolites to be synthesized by a hybrid NRPS-pteridine pathway. These studies expand our view of the combinatorial biosynthetic potential available in bacteria.

*For correspondence: jason. crawford@yale.edu

†These authors contributed equally to this work

Competing interests: The authors declare that no competing interests exist.

## Introduction

Nonribosomal peptides are a structurally and functionally privileged class of natural products constructed from a highly diverse pool of potential proteinogenic and nonproteinogenic amino acid building blocks (*Walsh et al., 2013*; *Walsh, 2016*). Members of the family include well-known pharmacologically relevant agents, such as vancomycin, daptomycin, penicillin, cyclosprin, and many others. The core catalytic domains of a minimal nonribosomal peptide synthetase (NRPS) extender module include condensation (C), adenylation (A), and peptidyl-carrier protein (PCP, a.k.a., thiolation, T) domains. The NRPS first selects its cognate amino acid from the available substrate pool, a step controlled by the selectivity of the A domain, activates it as an aminoacyl adenylate, and subsequently loads it onto the PCP domain. The C domain then typically catalyzes the formation of a *trans*-amide bond establishing individual peptide backbone linkages through nucleophilic attack of the free amino group present on a downstream aminoacyl-PCP on the upstream peptidyl-PCP. NRPSs can also engage in 'hybrid' pathways to dramatically expand their biocatalytic capabilities.

**eLife digest** Many bacteria produce small molecules that can be used as a basis for developing new drugs. The instructions for the pathways that make these molecules are stored in the genome – the complete set of genetic material – of the bacteria. With the advancements in genome sequencing over the last decade, these instructions are becoming much more readily available in sequence databases. "Genome mining" is a strategy that involves searching these databases to identify unknown biochemical pathways and help to characterize them. This strategy could help us to discover many more ecologically and medically relevant molecules.

A bacterium called *Photorhabdus luminescens* produces many antibiotics and other molecules that play a variety of roles in the bacterium's lifecycle. The bacteria live in the gut of roundworms, and the two species have a mutually beneficial relationship where they help each other to acquire food. However, the bacteria are less friendly to the insects that the roundworms infect. When *P. luminescens* is released into the body of an insect, it takes on a disease-causing form and releases toxic molecules that kill the insect.

Park, Perez et al. have now used genome mining to identify a biochemical pathway in *P. luminescens* that combines the pathways used to create two different types of small molecules produced by the bacteria. This "hybrid" pathway produces a new set of molecules – called pepteridines – that are released by the disease-causing form of the bacteria.

Park, Perez et al. also identified a regulator protein that controls the hybrid pathway. This regulator is known to help the bacteria to change into the form that kills insects. The pathway also affects the production of proteins known to be involved in "quorum sensing". In this process, bacteria use a diverse set of chemical signals to report how many other bacteria are nearby, which enables the bacteria to launch coordinated biological responses – for example, releasing toxic molecules – when their numbers are great enough.

In the future, further experiments will be pursued to rigorously characterize how the components of the new hybrid pathway work together. While the hybrid pathway responsible for the production of the pepteridines serves as one example of the utility of genome mining, there is still much room for further discovery. Applying a similar strategy to different organisms has the potential to uncover other pathways of biomedical relevance.

Of note are the NRPS-polyketide synthase (PKS) hybrid pathways, which produce important molecules, including rapamycin, bleomycin, epothilone, colibactin, and others.

In contrast to the 'assembly line' logic of the majority of NRPSs, pteridines, such as the cofactors biopterin and folate, are derived from guanosine triphosphate (GTP) (*Brown, 2006*). Pteridines are composed of fused pyrimidine and pyrazine rings, and natural pteridines are typically functionalized at the C-6 position of the pteridine core (*vide infra*). The electronic properties of this scaffold underlie its role in redox active cofactors critical to a host of metabolic transformations, such as hydroxylation of aromatic compounds and generation of the neurotransmitter nitric oxide (*Groehn et al., 2000*). Additionally, functionalized pteridines serve as reactants in a number of one-carbon group transfer reactions in metabolism and, more recently, were implicated in catalysis of 1,3-dipolar cyclo-addition-mediated decarboxylation reactions (*Payne et al., 2015*; *White et al., 2015*).

The increase in genomic sequence information from diverse microbial sources has highlighted enormous untapped metabolic potential for discovery of novel nonribosomal peptides and pteridines among other metabolite groups, including groups synthesized by new enzyme classes (*Cimermancic et al., 2014*). As a result of horizontal gene transfer events during evolution, it is known that the bacterial secondary metabolic pathways encoding many of these metabolites often reside on genomic islands (*Shankar et al., 2006*; *Penn et al., 2009*; *Ziemert et al., 2014*). Genome synteny analyses platforms enable *in silico* visualization of co-localized genetic loci among phylogenetically related organisms and represent general tools to aid in identifying genomic islands. We have previously used genome synteny analysis to aid in identification of 'atypical pathways' and to determine the biocatalytic functions of hypothetical proteins located on genomic islands (*Guo and Crawford, 2014*; *Vizcaino et al., 2014b*). Here, we employ genome synteny analysis to

identify a genomic island in the entomopathogen *Photorhabdus luminescens* TT01 harboring an unprecedented combination of nonribosomal peptide synthetase (NRPS)-like and pteridine synthase biosynthetic machineries, suggesting a novel type of hybrid pathway. We demonstrate that this pathway encodes new metabolites dependent on the hybrid enzymatic machinery, the pepteridines. *P. luminescens* is a Gram-negative Gammaproteobacterium that undergoes stochastic phenotypic variation, and through the use of genetically 'locked' variants (*Somvanshi et al., 2012*), we show that the pepteridines are produced in a specific variant associated with pathogenesis. Allelic-exchange mutagenesis in *P. luminescens* and comparative quantitative proteomic analysis further reveal that the genetic locus affects production of several groups of proteins related to the biosynthesis of known quorum sensing and secondary metabolite systems.

## Results and discussion

### Genome synteny illuminates a hybrid NRPS-pteridine genomic island

*P. luminescens* is a Gammaproteobacterium mutualistically associated with nematodes, and the pair prey on insect larvae (*Waterfield et al., 2009*). *P. luminescens* produces an assortment of bioactive molecules and antibiotics to regulate its mutualistic and pathogenic interactions (*Vizcaino et al., 2014b*; *Challinor and Bode, 2015*). Consequently, this genus, which includes one human pathogen, rivals the *Streptomyces* genus in terms of the number of secondary metabolic pathways in a given genome (*Duchaud et al., 2003*; *Tobias et al., 2016*). Using the MicroScope bioinformatics platform, we identified a genomic island (*plu2792-plu2799*, *Figure 1—figure supplement 1*), harboring mixed NRPS-pteridine synthase machinery (*Figure 1*). Protein sequence homology analysis demonstrated predicted pteridine biosynthetic enzymes, such as GTP cyclohydrolase (GTPCH) I, 6-pyruvoyltetrahydropterin synthase, pteridine reductase, and pteridine pyrophosphokinase. Interestingly, a NRPS carrier protein (thiolation domain, T) and a condensation domain (C) were genetically fused to a pyruvate dehydrogenase E2-like subunit (*plu2796*). The pyruvate dehydrogenase complex (E1, E2, and E3 subunits) is a well-studied set of enzymes that converts pyruvate to acetyl-CoA (*Patel and Roche, 1990*), serving as a key metabolic bridge between glycolysis and the citric acid cycle. A complementary E1-like subunit is also encoded in the pathway. This pathway was not detected by early versions of the antiSMASH algorithm (*Blin et al., 2013*) used for identifying the biosynthetic pathways of secondary metabolites; however, recent integration of the ClusterFinder algorithm which aids in identifying divergent biosynthetic gene clusters of unknown classes (*Cimermancic et al., 2014*; *Weber et al., 2015*) allowed detection of part of the pathway (*plu2796-plu2798*) as a likely pathway of secondary metabolic enzymes.

### Heterologous expression of the pepteridine biosynthetic pathway

We first cloned the hybrid NRPS-pteridine genomic island without its clustered regulatory protein (Plu2792) and placed it under the control of a phage T7 promoter for heterologous expression in *Escherichia coli* BAP1 (*Pfeifer et al., 2001*). Comparison of the culture broths from *E. coli* harboring the pathway relative to that of an empty vector control (pET28a) revealed a pathway-dependent yellow phenotype (*Figure 2B*). Initial comparative single quadrupole liquid chromatography-mass

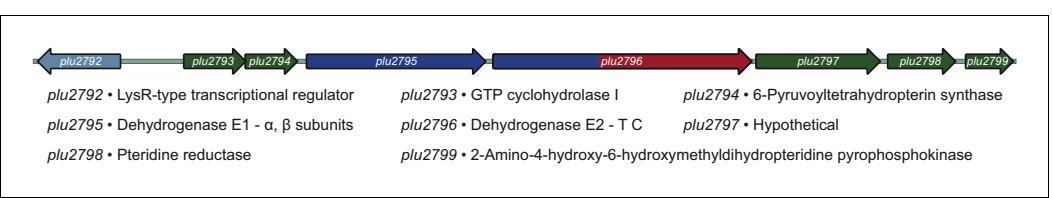

**Figure 1.** The pepteridine biosynthetic locus. Green, pteridine synthesis genes; Blue, pyruvate dehydrogenase-like genes; Red, NRPS-like genes. T, thiolation domain; C, condensation domain.
The following figure supplement is available for figure 1:

**Figure supplement 1.** Genome synteny analysis using MicroScope.

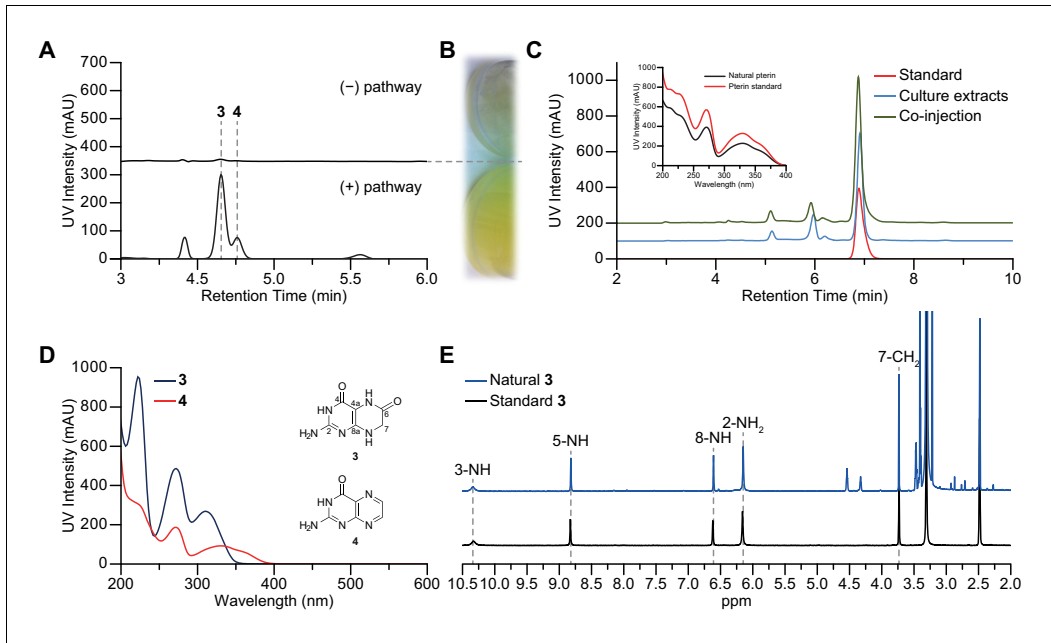

**Figure 2.** Production of 7,8-dihydroxanthopterin (**3**) and pterin (**4**) by heterologous expression. (**A**) HPLC traces (310 nm) of butanol extracts showing compounds **3** and **4** being over-produced in the heterologous expression strain. (**B**) The edge of a culture flask growing *E. coli* BAP1 carrying an empty vector pET28a (top), and a culture flask growing *E. coli* BAP1 carrying the wild-type pepteridine pathway (bottom). (**C**) HPLC traces (310 nm) from HPLC co-injection with authentic pterin (**4**), and UV absorption spectral comparison between natural and authentic pterin (inset). (**D**) UV-vis absorption spectra of compounds **3** and **4**. (**E**) $^1$H NMR comparison of natural (blue) and standard (black) **3** in DMSO-$d_6$.

The following figure supplements are available for figure 2:

**Figure supplement 1.** $^1$H and $^{13}$C NMR spectra of compound **3**.

**Figure supplement 2.** 2D- (gCOSY and gHMBCAD) NMR spectra of compound **3**.

spectrometry (LC/MS) analysis of their butanol extracts revealed two dramatically increased peaks in the broth harboring the pathway (*Figure 2A*). Through extensive 1D- and 2D-NMR, high-resolution-electrospray ionization-quadrupole-time-of-flight-mass spectrometry (HR-ESI-QTOF-MS), and LC/MS co-injections with authentic standards (*Figure 2C–E* and *Figure 2—figure supplements 1 and 2*), the chemical structures of the small molecules attributed to these peaks were assigned as known 2-amino-4-oxopteridine small molecules, 7,8-dihydroxanthopterin (**3**) and pterin (**4**).

## Comparative metabolomic analysis and pathway-targeted molecular networking

To identify novel lower abundance advanced metabolites dependent on both the NRPS and pteridine biosynthetic enzymes, we individually constructed nonpolar genetic deletions of every biosynthetic gene in the pathway, maintaining transcriptional and translational control elements for comparative metabolomics. Relative to the control vector, we identified 224 molecular features in butanol extracts that were dependent on the presence of the wild-type pathway (*Supplementary file 1A*). These include small molecules that are encoded by the pathway, shunt metabolites emerging from the pathway, and host metabolites that are enhanced by the pathway from an undetectable to a statistically significant level. Of these, 114 were dependent on the GTPCH I homolog, Plu2793, which is predicted to convert GTP to 7,8-dihydroneopterin triphosphate, initiating pteridine synthesis (*Burg and Brown, 1968*). The predicted functional redundancy with primary metabolic enzymes is expected to account for the larger number of metabolic perturbations

observed for *plu2793*. Thirty-seven molecular features were dependent on the atypical dehydrogenase E2-NRPS fusion enzyme, Plu2796, and only 12 were dependent on both. The number of wild-type molecular features that were dependent on the remaining biosynthetic enzymes, Plu2794, Plu2795, Plu2797, Plu2798, and Plu2799, are listed in *Table 1*.

We then conducted tandem MS analysis on the pathway-dependent molecular features for pathway-targeted molecular networking (*Vizcaino et al., 2014a*). Molecular networking is a powerful approach to network molecules based on tandem MS fragmentation similarities (*Watrous et al., 2012*). Pathway-targeted molecular networking focuses on networking of metabolites dependent on the presence of a functional pathway (e.g., wild-type versus a secondary metabolic pathway mutant). By assessing and mapping the relative production levels of the metabolites onto the larger wild-type pepteridine network (*Figure 3—figure supplements 1* and *2*), we could visualize how individual genetic mutations affect overall metabolite distributions at a systems level (*Figure 3*). White nodes represent metabolites abolished in a given mutant strain, thereby dramatically focusing metabolite discovery efforts. Some deletions led to a decrease in, rather than complete abolishment of, select molecular feature production, indicating that substantial metabolic crosstalk occurs between this specific hybrid secondary metabolic pathway and primary metabolism. With the observed crosstalk, we focused our structural characterization efforts on the smaller number of molecular features that were dependent on the atypical E2-NRPS machinery. We also included pterin (**4**, green node in *Figure 3*) as a standard in our network generation to aid in defining metabolites possessing pteridine structural scaffolds. Through this analysis, two prominent metabolites were selected for NMR-based structural characterization (*Figure 3*).

## Structural characterization of the pepteridines

Compounds **1** and **2** with molecular ions at $m/z$ 224.1141 ([M+H]$^+$, calcd $m/z$ 224.1147, $C_9H_{14}N_5O_2$) and $m/z$ 210.0987 ([M+H]$^+$, calcd $m/z$ 210.0991, $C_8H_{12}N_5O_2$), respectively, appeared to be structurally related in the molecular network with a 14 Da mass difference likely attributable to $CH_2$. A bacterial culture in M9-minimal medium (6 l) supplemented with casamino acids (5 g/l) was initiated for structural characterization. LC/MS analysis of the accumulated butanol extracts led to detection of compounds **1** ($t_R$ = 7.3 min) and **2** ($t_R$ = 5.8 min) (*Figure 3—figure supplement 3*). However, initial isolation attempts proved challenging because of the low solubility and high polarity of these molecules, which co-eluted with the amino acid supplements. Consequently, isolation of **1** and **2** proceeded through larger scale cultivation in M9-minimal medium (32 l). Although a much lower yield was observed, the isolation was streamlined in the absence of amino acid supplements. The clarified medium was lyophilized, and the dried residue was extracted with 50% aqueous methanol (2 l). Flash column chromatography ($C_{18}$) followed by several rounds of reverse-phase liquid chromatography separation led to isolation of pure compounds **1** (0.8 mg) and **2** (1.2 mg), which we named pepteridines A and B, respectively.

**Table 1.** Genetic dependency of molecular features.

| Gene | Dependent molecular features |
| --- | --- |
| *plu2793* | 114 |
| *plu2794* | 31 |
| *plu2795* | 19 |
| *plu2796* | 37 |
| *plu2797* | 10 |
| *plu2798* | 21 |
| *plu2799* | 26 |
| *plu2793/plu2796* | 12 |

Of the wild-type molecular features, this table denotes how many of them were deleted (i.e., not detected) in each mutant strain.

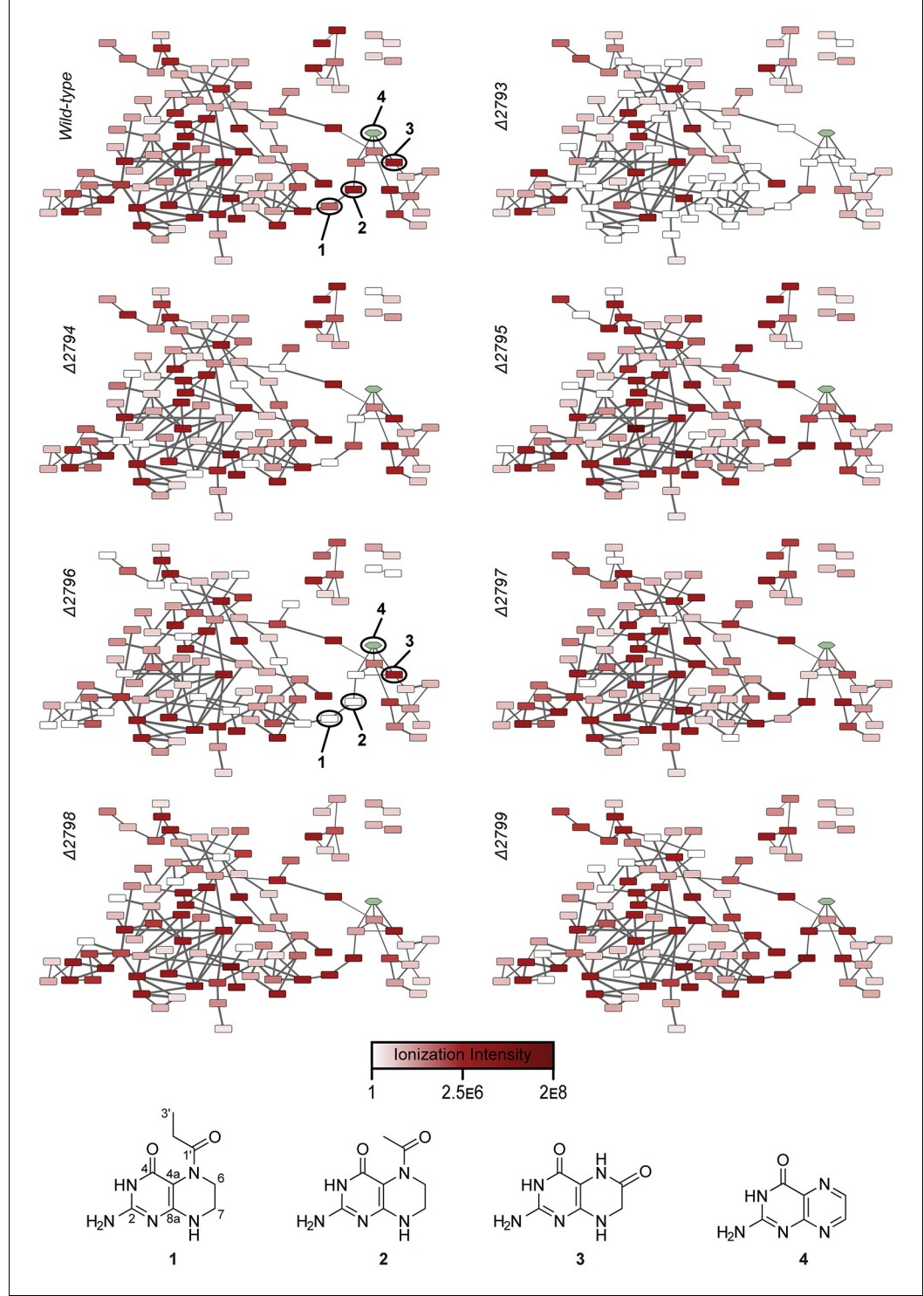

**Figure 3.** Relative abundances of wild-type pathway-dependent metabolites in wild-type and mutant pepteridine pathways. The average ionization intensity is depicted for each molecular feature under a given genetic condition (wild-type and *Δplu2793* through *Δplu2799*). Alterations in abundance are correlated with changes in nodal color intensity among genetic constructs, allowing visual assessment of product distributions for a given mutation. See *Figure 3—figure supplement 2* for information on mass of pathway-dependent metabolites. Structural characterization of compounds **1** and **2** is shown in *Figure 3—figure supplements 4–10*, *Table 2*, and the Materials and methods section.

*Figure 3 continued on next page*

*Figure 3 continued*

The following figure supplements are available for figure 3:

**Figure supplement 1.** Untargeted molecular networking of culture extracts from cells harboring the wild-type pepteridine biosynthetic pathway.

**Figure supplement 2.** Pathway-targeted molecular networking of the wild-type pepteridine biosynthetic pathway.

**Figure supplement 3.** LC/MS Extracted Ion Count (EIC) chromatograms of compounds **1** (A) and **2** (B) from butanol extracts of the pepteridine heterologous expression strain.

**Figure supplement 4.** $^1$H and $^{13}$C NMR spectra of compound **1**.

**Figure supplement 5.** 2D- (gCOSY and gHSQCAD) NMR spectra of compound **1**.

**Figure supplement 6.** 2D- (gHMBCAD and NOESY) NMR spectra of compound **1**.

**Figure supplement 7.** $^1$H and $^{13}$C NMR spectra of compound **2**.

**Figure supplement 8.** 2D- (gCOSY and gHSQCAD) NMR spectra of compound **2**.

**Figure supplement 9.** 2D- (gHMBCAD and NOESY) NMR spectra of compound **2**.

**Figure supplement 10.** Amide conformation of compounds **1** and **2**.

---

Structural elucidation of pepteridine A (**1**) and B (**2**) was achieved through interpretation of 1D- ($^1$H and $^{13}$C) and 2D-NMR (gCOSY, gHSQCAD, gHMBCAD, and NOESY) spectral data (*Figure 3— figure supplements 4–9* and *Table 2*). Briefly, $^1$H NMR spectral data coupled with gHSQCAD of **1** were used to show that three NH or $NH_2$ exchangeable protons, three methylene protons, and one methyl signal were present, suggesting that compound **1** is composed of one hydrogenated pteri-dine scaffold and one distinguished acyl group. Interpretation of gCOSY and gHMBCAD NMR spec-tra established the two partial structures to be 2-amino-5,6,7,8-tetrahydropteridin-4(3*H*)-one and a propionyl group. Key HMBC correlations from the propionyl protons and the methylene protons at C-6 in the pteridine scaffold demonstrated the sharing of an amide carbonyl. These correlations established the connectivity of the propionyl group at N-5, thereby characterizing **1** as a 2-amino-5-propionyl-5,6,7,8-tetrahydropteridin-4(3*H*)-one (*Figure 3*). Interpretation of NOESY NMR data and comparison of the proton chemical shifts at the pteridine C-6 to *cis*- and *trans*-amide analogs sup-ported the *cis*-amide conformation of the acyl group at N-5 in solution (*Figure 3—figure supple-ment 10*) (*Lanyon-Hogg et al., 2015*). The critical difference between **1** and **2** was the absence of a $CH_2$ signal in the NMR spectral data, supporting the assumption that **2** contains an acetyl group in place of the propionyl group in **1**, and further HMBC NMR analysis and HR-ESI-QTOF-MS data con-firmed this.

## Delineating the biosynthesis of pepteridine A and B

Combining the bioinformatic, genetic, and comparative metabolomic analyses with the new struc-tures suggests a hybrid biosynthetic route to the pepteridines, consisting of pteridine synthesis from GTP, acyl-synthesis via oxidative decarboxylation of α-keto acids, and NRPS-dependent condensa-tion of these distinct substrates (*Figure 4*). The presence of predicted GTPCH I, 6-pyruvoyltetrahy-dropterin synthase, and pteridine reductase homologs in the pathway supports the formation and redox control of a tetrahydropterin substrate. The presence of the pyruvate-dehydrogenase-like E1 and E2-NRPS fusion enzymes supports derivation of the pepteridine acyl-appendages from α-keto acids through an analogous dehydrogenase mechanism. It is likely, however, that the pathway inter-acts with other pteridine and dehydrogenase biosynthetic enzymes encoded in the genome of the heterologous host, *E. coli*, such as the E3 subunit of the pyruvate dehydrogenase complex, which is required for lipoamide regeneration but absent from the pathway. Rather than producing an acyl-

**Table 2.** NMR spectral data of pepteridine A (**1**) and B (**2**) in DMSO-$d_6$.

**Pepteridine A (1)**

| No. | $\delta_C{}^a$ | type | $\delta_H{}^b$ | Mult (J in Hz) | HMBC |
|---|---|---|---|---|---|
| 1 | | N | | | |
| 2 | 154.6 | C | | | |
| 3 | | NH | 10.07 | br s | |
| 4 | 157.3 | C | | | |
| 4a | 93.1 | C | | | |
| 5 | | N | | | |
| 6 | 38.5 | CH$_2$ | 4.54 | dd (12.2, 3.6) | C-1', C-4a |
| | | | 2.33 | m | |
| 7 | 41.9 | CH$_2$ | 3.30 | d (12.2) | |
| | | | 2.97 | dt (12.0, 4.2) | C-6 |
| 8 | | NH | 6.96 | br s | C-4a, C-6, C-7 |
| 8a | 153.1 | C | | | |
| 9 | | N | | | |
| 1' | 174.0 | C | | | |
| 2' | 26.5 | CH$_2$ | 2.57 | dt (15.0, 7.4) | C-1', C-3' |
| | | | 2.15 | dt (14.8, 7.3) | C-1', C-3' |
| 3' | 9.7 | CH$_3$ | 0.88 | t (7.4) | C-1', C-2' |
| | | NH$_2$ | 6.25 | br s | |

**Pepteridine B (2)**

| No. | $\delta_C{}^a$ | type | $\delta_H{}^b$ | Mult (J in Hz) | HMBC |
|---|---|---|---|---|---|
| 1 | | N | | | |
| 2 | 155.2 | C | | | |
| 3 | | NH | 10.04 | br s | |
| 4 | 157.3 | C | | | |
| 4a | 93.5 | C | | | |
| 5 | | N | | | |
| 6 | 38.3 | CH$_2$ | 4.52 | dd (12.1, 3.5) | C-1', C-4a |
| | | | 2.32 | dt (11.7, 2.4) | |
| 7 | 41.8 | CH$_2$ | 3.30 | d (12.0) | |
| | | | 2.98 | dt (11.9, 4.1) | C-6 |
| 8 | | NH | 6.96 | d (4.0) | |
| 8a | 153.3 | C | | | |
| 9 | | N | | | |
| 1' | 170.6 | C | | | |
| 2' | 22.5 | CH$_3$ | 1.97 | s | C-1' |
| | | NH$_2$ | 6.22 | br s | |

NMR spectra were recorded at $^b$600 MHz for $^1$H NMR and $^a$100 MHz for $^{13}$C NMR, respectively.

CoA, as in the pyruvate dehydrogenase reaction, the CoA-derived carrier protein arm (T-domain) could be directly primed in place of CoA (*Figure 4*). Analogous carrier protein priming mechanisms have been proposed for branched chain fatty acid substrate utilization in the formation of *N*-acyl-amides and the pristinamycin IIa streptogramin antibiotic (*Craig and Brady, 2011*; *Brachmann et al., 2012*); and for a glycolicacyl-NRPS extender unit in formation of the

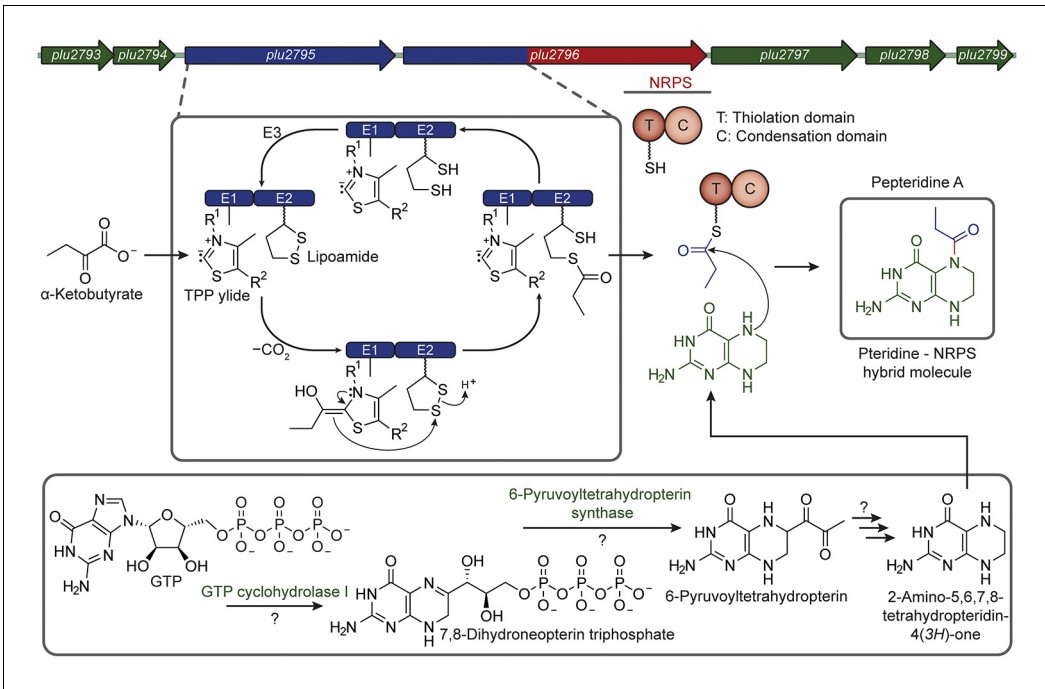

**Figure 4.** Proposed pepteridine biosynthesis. α-Ketobutyrate is processed by the atypical dehydrogenase-NRPS-like complex to generate a propionyl-loaded carrier protein (blue). Pteridine enzymes interact with the metabolism of the host to generate tetrahydropterin (green). The NRPS C domain couples these substrates to form the *cis*-amide bond (red) as illustrated for **1**.

naphthyridinomycin antitumor antibiotic (*Peng et al., 2012*). In contrast to these pathways, we propose that the resulting loaded acyl-carrier protein in pepteridine biosynthesis would be unusually condensed with a free tetrahydropterin substrate by the atypical NRPS C domain to install the *cis*-amide acyl-linkage. It is currently unclear whether the C domain catalyzes direct *cis*-amide bond formation or whether a *trans*-amide is formed and then isomerized to the observed *cis*-conformation in the pepteridines.

To gain additional support for our proposed biosynthesis, we analyzed pepteridine production in our full series of pathway mutant strains. Differential LC/HR-ESI-QTOF-MS analysis of butanol extracts demonstrated that pepteridine A (**1**) detection was completely abolished in the GTPCH I homolog mutant (Δ*plu2793*), the 6-pyruvoyl-tetrahydropterin synthase homolog mutant (Δ*plu2794*), and the atypical E2-NRPS mutant (Δ*plu2796*), indicating a genetic requirement for both pteridine and NRPS biosynthetic machineries (*Figure 5A*). Production of pepteridine B (**2**) was substantially reduced in the Δ*plu2793* and Δ*plu2794* strains, and completely abolished in the Δ*plu2796* mutant strain (*Figure 5C*). The minor residual production of **2** in these specific pteridine knockout strains is consistent with our observation that pterin (**4**) substrates can be detected at low abundance in the control strain (pET28a), further highlighting the metabolic crosstalk between this pathway and primary metabolism. To further demonstrate the biosynthetic dependence on the NRPS machinery, a Plu2796 S434A point mutant was generated in the wild-type pathway construct to site-specifically inactivate the NRPS carrier protein domain. Assessment of pepteridine production from this construct in comparison with wild-type, Δ*plu2796*, and control constructs confirmed dependency on the NRPS machinery (*Figure 5B,D*).

We then supplemented bacterial cultures harboring the wild-type pathway with varying concentrations of free α-ketobutyrate and pyruvate and, as expected, observed enhanced production of the respective pepteridines in a dose-dependent manner (*Figure 6A,B* and *Figure 6—figure supplement 1*). Additionally, universally $^{13}C$-labeled α-ketobutyrate ($^{13}C_4$) supplementation led to $^{13}C_3$-labeling of pepteridine A (**1**), as determined by HR-ESI-QTOF-MS (m/z 227.1244 ([M+H]$^+$, calcd m/z 227.1248, $^{13}C_3{}^{12}C_6H_{14}N_5O_2$), further supporting the proposed biosynthesis (*Figure 6C* and

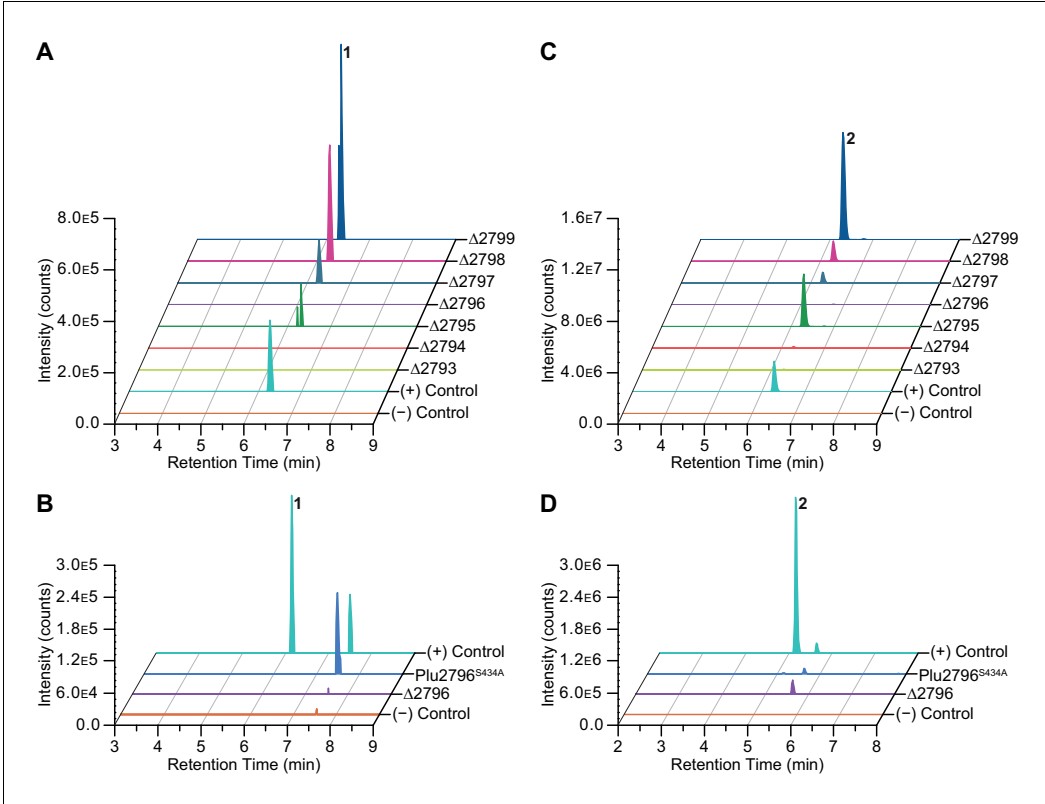

**Figure 5.** Gene deletion and NRPS inactivation analyses on **1** (A–B) and **2** (C–D). (A–B) Extracted ion chromatograms of **1** for wild-type and mutants are shown. (C–D) Extracted ion chromatograms of **2** for wild-type and mutants are shown.

*Figure 6—figure supplement 2*) and providing a forward path for protein biochemistry studies on this unprecedented enzymatic system. Importantly, these studies link pepteridine production to α-ketoacid substrate availability (*i.e.*, pyruvate and α-ketobutyrate) rather than free acyl-CoA substrates (*i.e.*, acetyl- and propionyl-CoA).

## Pepteridines are produced in the P-form phenotypic variant of *P. luminescens*

*Photorhabdus* bacteria are both pathogens to insects and mutualists to a specific nematode host (*Clarke, 2008*; *Waterfield et al., 2009*; *Clarke, 2014*). *Photorhabdus asymbiotica* can also cause infections in humans (*Gerrard et al., 2004*). To enhance their fitness for these variable host-bacteria objectives, *Photorhabdus* bacteria undergo phenotypic variation, which is controlled by a stochastic invertible promoter switch (*Somvanshi et al., 2012*). The orientation of the promoter regulates the formation of P- and M-form phenotypic variants. The P-form, typically the dominant variant in wild-type cultures, is pathogenic to insects. The P-form switches to the M-form, a small colony variant, which adheres to specific cells in the nematode intestine. It is thought that the M-form phenotype participates in colonization of its mutualistic nematode host. To determine whether the pepteridines are produced in *P. luminescens* and in which variant, we analyzed butanol extracts of *P. luminescens* genetically 'locked' in the M- and P-forms using LC/HR-ESI-QTOF-MS. In the M9-base medium, both pepteridines A and B could be detected in the pathogenic P-form phenotypic variant (*Figure 7*). However, under identical conditions, no production was observed in the M-form. These studies link pepteridine structure to phenotypic variant status and suggest that the pepteridines may participate in P-form biological activities.

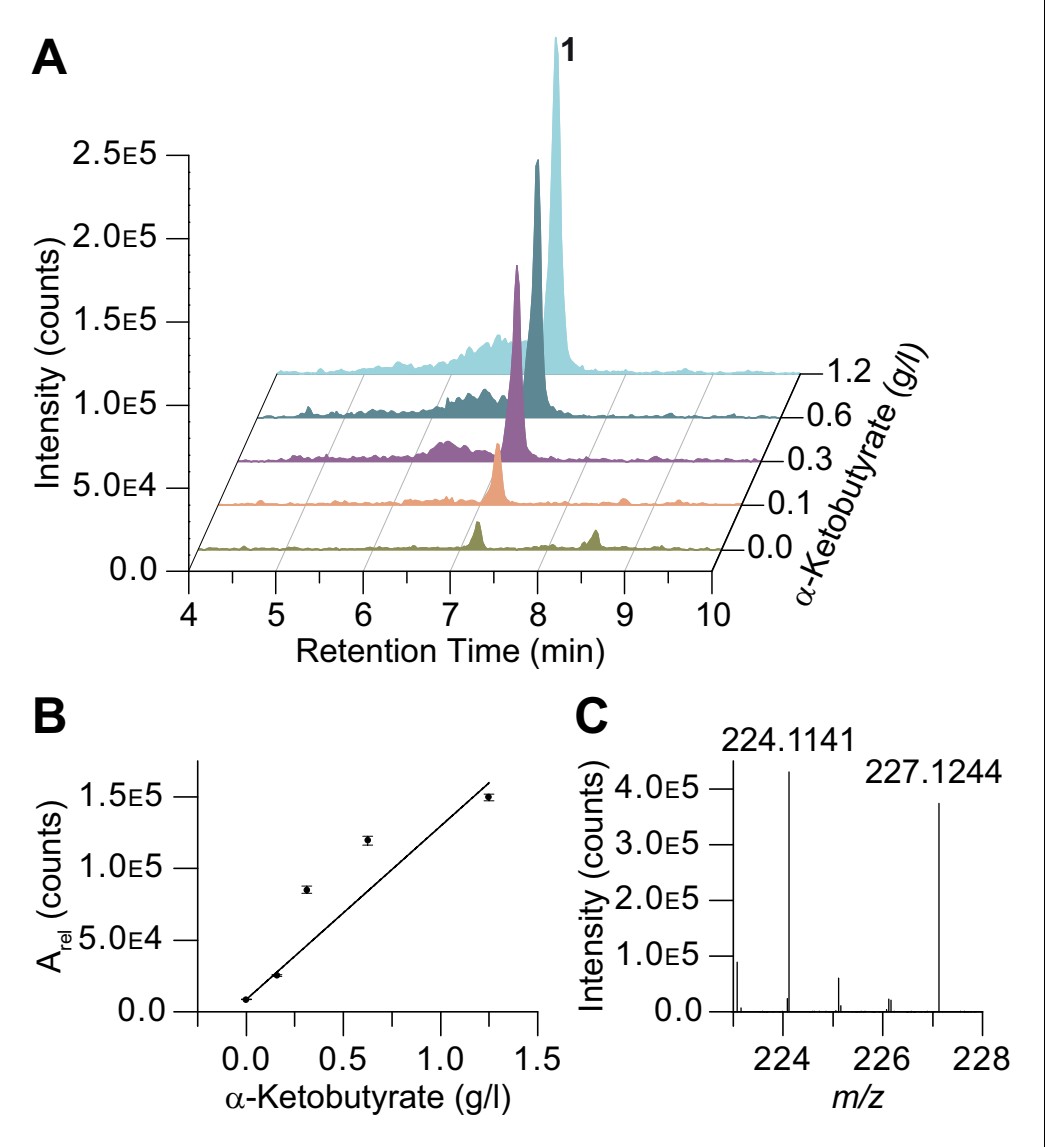

**Figure 6.** α-Ketobutyrate feeding studies on **1**. (A–B) Production enhancement of **1** in α-ketobutyrate supplementation studies ($A_{rel}$, relative integration value). (C) $^{13}C_4$-α-ketobutyrate supplementation leads to $^{13}C_3$ incorporation in **1** consistent with the proposed biosynthesis.

The following figure supplements are available for figure 6:

**Figure supplement 1.** Production of compounds **1** and **2** in dose-dependent α-ketobutyrate (**A**) and pyruvate (**B**) feeding studies.

**Figure supplement 2.** Characterization of $^{13}C_4$-α-ketobutyrate incorporation in compound **1**.

## The pepteridine genetic locus affects protein production in quorum sensing and secondary metabolism

To initiate functional cellular studies, we deleted the pepteridine genetic locus in *P. luminescens* TT01 on a wild-type background (*Δlocus*) and a *ΔhexA* background (*ΔhexA/locus*) using allelic-exchange mutagenesis. HexA is a LysR-type transcriptional repressor that regulates stilbene production and participates in the insect pathogen, nematode mutualist transition (*Joyce and Clarke, 2003*; *Kontnik et al., 2010*), and global transcriptomic analysis indicates that pepteridine genes are

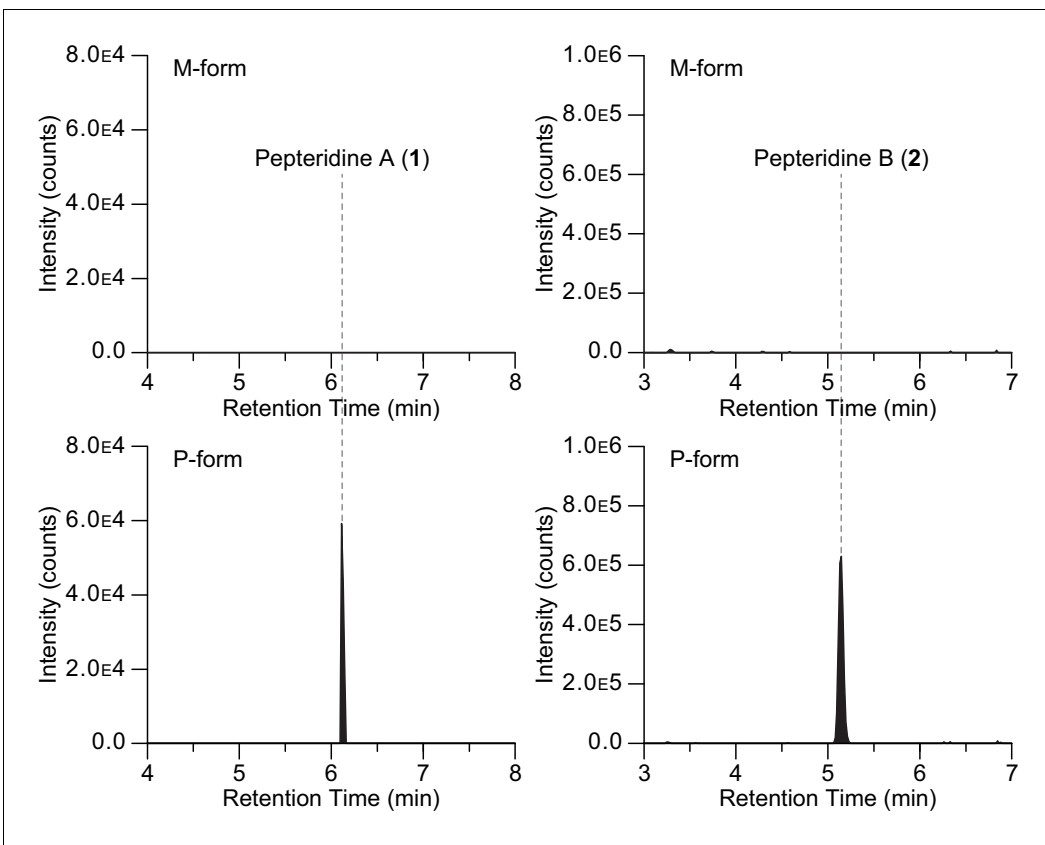

**Figure 7.** Extracted ion counts LC/HR-ESI-QTOF-MS analysis of pepteridines A (left panel) and B (right panel) from two genetically engineered *P. luminescens* strains locked in the phenotypic variants M-form (top) and P-form (bottom).

The following figure supplement is available for figure 7:

**Figure supplement 1.** Extracted ion counts chromatograms from LC/HR-ESI-QTOF-MS analysis of pepteridines A (left panel) and B (right panel) from the butanol extracts of P-form culture broth (top), standard compounds (middle), and co-injection (bottom).

upregulated in *ΔhexA/hfq* strains (*Tobias et al., 2016*). We cultivated the wild-type and mutant strains in a culture medium based on the high concentrations of free proteinogenic amino acids found in insect hemolymph (*Crawford et al., 2010*). Twenty-four hour cultures were centrifuged, the cells were rapidly lysed, and the protein fractions were trypsinized for quantitative proteomic analysis. Relative fold changes (wild-type vs *Δlocus* and *ΔhexA* vs *ΔhexA/locus*) were calculated based on quantitative LC/MS-MS analyses (*Figure 8*). On a wild-type background, deletion of the locus had little effect on the proteome (*Figure 8A*). However, on a *ΔhexA* background and consistent with the transcriptomic data, more dramatic proteomic effects were observed, supporting regulation of the pepteridine genomic locus by HexA (*Figure 8B*). Proteins that participate in pyrone quorum sensing and secondary metabolism were downregulated in the *ΔhexA/locus* mutant relative to the *ΔhexA* control. For example, deletion of the pepteridine genetic locus led to a 5-fold decrease of the enzyme Plu4844 in the *ΔhexA* background (*Figure 8B*). Plu4844 participates in synthesis of pyrone autoinducers (*Brachmann et al., 2013*). We similarly observed a 3–4-fold decrease of the enzymes Plu2817, Plu2204, and Plu4187. These enzymes are involved in biosynthesis of the potent phenoloxidase inhibitor rhabduscin (Plu2817) (*Crawford et al., 2010*, *2012*); cinnamic acid (Plu2204, 2207, 2208), a key substrate of the multipotent stilbenes (*Joyce et al., 2008*); and the polyketide anthraquinone pigments (Plu4186, 4187, 4188, 4192) (*Brachmann et al., 2007*). Collectively, our proteomic

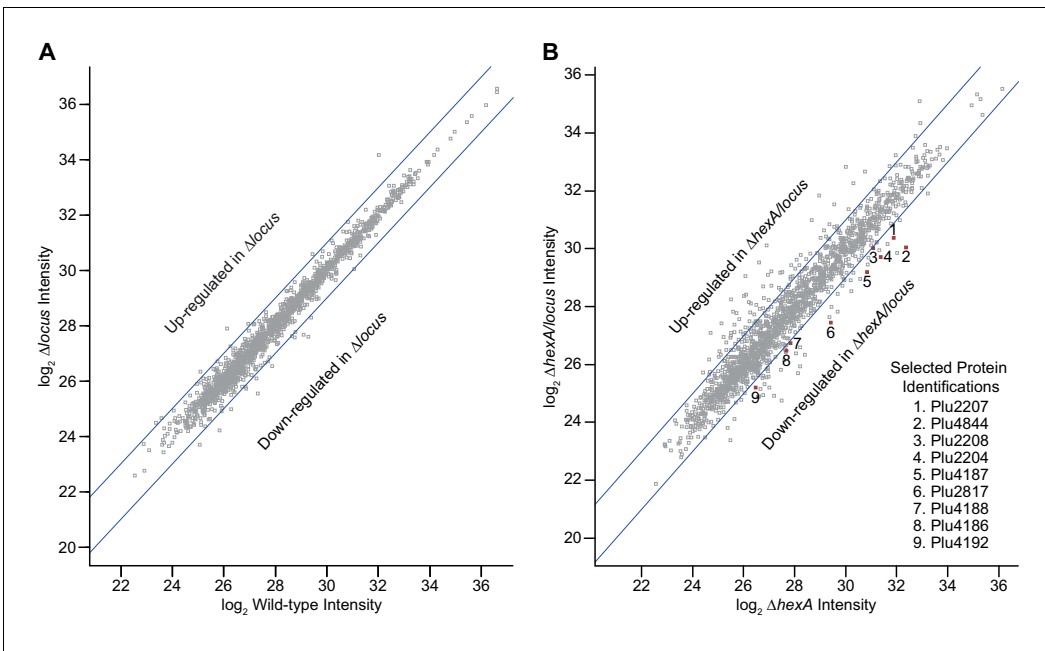

**Figure 8.** Quantitative proteomic analysis of a *Δlocus* strain in a wild-type background (A) and *ΔhexA* background (B).

The following source data is available for figure 8:

**Source data 1.** Proteins increased in WT vs. WT*Δlocus* strains by LC-MS/MS.
**Source data 2.** Proteins increased in WT*Δlocus* vs. WT strains by LC-MS/MS.
**Source data 3.** Proteins increased in *ΔhexA* vs. *ΔhexAΔlocus* strains by LC-MS/MS.
**Source data 4.** Proteins increased in *ΔhexAΔlocus* vs. *ΔhexA* strains by LC-MS/MS.
**Source data 5.** All proteins observed in WT and WT*Δlocus* strains by LC-MS/MS.
**Source data 6.** All proteins observed in *ΔhexA* and *ΔhexAΔlocus* strains by LC-MS/MS.

data support that the pepteridine genomic locus is regulated by HexA and positively affects pyrone quorum sensing and select secondary metabolic pathways.

## Conclusion

Bacterial natural products represent a rich source of lead structures in small molecule drug discovery efforts and serve as excellent molecular probes in biology. Novel classes of putative biosynthetic enzymes, including enzymes belonging to new types of "hybrid" metabolic pathways, can be identified in expanding genome sequence databases through genome mining approaches. Integration of two or more distinct types of metabolic pathways provides combinatorial biosynthetic routes to expand the structural and functional properties of natural products. In this study, using genome synteny analysis, we identified an unprecedented genomic island in *P. luminescens* that harbored both NRPS-like and pteridine biosynthetic machineries. This hybrid biosynthetic pathway is responsible for synthesis of new pteridine metabolites, the pepteridines. Pepteridine A (**1**) and B (**2**), which respectively harbor atypical *cis*-amide functionalized $C_3$ and $C_2$ acyl-chains off a free pterin substrate, a particularly unusual feature for a NRPS, were extensively characterized by NMR. The genetic determinants of pepteridine production were assessed in a heterologous host and networked at the systems level, providing global pathway-dependent maps to visualize relative metabolite distributions among wild-type and mutant strains. The pepteridine metabolites were also identified in the wild-type host, *P. luminescens*, specifically associated with its phenotypic variant linked to pathogenesis

(the P-form). Quantitative proteomic analysis further demonstrated that when the LysR-type transcriptional repressor is derepressed (*ΔhexA*), the pepteridine genetic locus positively affects pyrone qurorum sensing and select secondary metabolic pathways (*i.e.*, these pathways were downregulated in the *ΔhexA/locus* strain relative to the *ΔhexA* strain). The pepteridines likely participate in chemical signaling. Indeed, in the Gram-negative plant pathogen *Agrobacterium tumefaciens*, pterin signaling was recently implicated in biofilm regulation (*Feirer et al., 2015*). The pepteridines discovered here by targeting atypical genomic sequence space, to the best of our knowledge, represent the first metabolites to be characterized from a hybrid NRPS-pteridine biosynthetic gene cluster. Characterization of these types of atypical hybrid pathways expands our view on the combinatorial biosynthetic potential available in nature's metabolic toolbox.

## Materials and methods

### Cloning of the pepteridine pathway

The genomic island spanning genes *plu2793* to *plu2799* from *Photorhabdus luminescens* TT01 were amplified via PCR using Phusion High-Fidelity DNA Polymerase (New England Biolabs (NEB), USA) and primers *2796-cluster-5* and *2796-cluster-3* (*Supplementary file 1B*). Reactions (50 µl) were assembled according to the manufacturer's protocol with inclusion of 5% (v/v) dimethyl sulfoxide and the use of 1 µl of confluent *P. luminescens* TT01 as template. Thermal cycling was carried out on a C1000 Touch thermal cycler equipped with a Dual 48/48 Fast Reaction Module (Bio-Rad, USA). Amplification was assessed using a 0.75% agarose gel in $1 \times$ TAE (40 mM Tris pH 7.6, 20 mM Acetic Acid, and 1 mM EDTA) stained with GelGreen (Biotium, USA). Amplification products were purified using the QIAquick PCR Purification Kit (Qiagen, USA) according to the manufacturer's protocol. The purified product and pET24b (EMDMillipore - Novagen, USA) were individually digested with NdeI and XhoI (NEB), and purified using the QIAquick PCR Purification Kit. Ligation of the products was carried out using T4 DNA ligase (NEB). Ligation mixtures were directly transformed into 50 µl of MAX Efficiency DH5α chemically competent cells (Invitrogen, USA) and recovered in 200 µl super optimal broth with catabolite repression (SOC, 2% (w/v) tryptone, 0.5% (w/v) yeast extract, 10 mM NaCl, 2.5 mM KCl, 10 mM $MgCl_2$, 10 mM $MgSO_4$, and 20 mM glucose) following the manufacturer's protocol. Successful transformants were selected by plating 150 µl of the transformation outgrowth onto lysogeny broth (LB) agar (BD, USA; 1% (w/v) tryptone, 0.5% (w/v) yeast extract, 1% (w/v) NaCl, and 1.5% (w/v) agar) plates supplemented with 25 µg/ml kanamycin (American Bioanalytical, USA) and overnight growth at 37°C. Single, well-defined colonies were picked and grown overnight as suspension cultures at 37°C and 250 rpm in 5 ml LB (BD; 1% (w/v) tryptone, 0.5% (w/v) yeast extract, and 1% (w/v) NaCl) supplemented with 25 µg/ml kanamycin. Plasmids were harvested using the QIAprep Spin Miniprep Kit (Qiagen) according to the manufacturer's protocol. Incorporation of the biosynthetic pathway encompassing genes *plu2793-plu2799* was assessed by end sequencing using the T7 promoter and T7 terminator primers (Genewiz, USA; or Keck Foundation Biotechnology Resource Laboratory at Yale University, USA). Primer walking fully validated the inserted genetic sequence along with the plasmid's transcriptional and translational control elements proximal to the region of insertion. The construct was named pEplu2796. Sequencing primers are listed in *Supplementary file 1C*.

### Preparation of constructs for heterologous expression studies

*Escherichia coli* BAP1 (*Pfeifer et al., 2001*) chemically competent cells were prepared according to standard molecular biology protocols and transformed using 1 µl of the desired construct. Transformants were selected via overnight growth at 37°C on LB agar plates supplemented with 25 µg/ml kanamycin. Resulting single colonies were picked for overnight liquid culture at 37°C and 250 rpm in LB media supplemented with 25 µg/ml kanamycin. Glycerol stocks were subsequently prepared for long-term storage.

### Small-scale growth and comparative chemical analysis

The *E. coli* BAP1 strain carrying the selected pathway was spread onto an LB agar plate supplemented with 25 µg/ml kanamycin and incubated at 37°C overnight. A single colony was inoculated into 5 ml LB liquid media containing 25 µg/ml kanamycin and grown overnight at 37°C and 250 rpm.

5 ml of M9 minimal medium (Amresco, USA; 0.6% (w/v) $Na_2HPO_4$, 0.3% (w/v) $KH_2PO_4$, 0.05% (w/v) NaCl, 0.1% (w/v) $NH_4Cl$, 0.2% (w/v) glucose, 2 mM $MgSO_4$, and 0.1 mM $CaCl_2$) supplemented with 5 g/l casamino acids (Amresco) and 25 µg/ml kanamycin was inoculated at 1:200 with the overnight growth, and grown at 37°C and 250 rpm until the optical density ($OD_{600}$) reached 0.5–0.6. Growth was slowed by placing the culture on ice. Production was induced via the addition of isopropyl-1-thio-$\beta$-D-galactopyranoside (IPTG, American Bioanalytical) at a final concentration of 0.1 mM and growth proceeded at 25°C and 250 rpm. After 72 hr, the culture was centrifuged (2000 $\times$ g, 20 min, 4°C) and the supernatant was then extracted 1:1 with water-saturated butanol (1 $\times$ 5 ml). The organic fraction was dried under reduced pressure on a Genevac (USA) HT-4X evaporation system for 2 hr. The crude material, which was re-suspended in 100 µl of 50% aqueous methanol, was analyzed via single quadrupole LC/MS (Column: Phenomenex Kinetex $C_{18}$ (100 Å) 5 µm (4.6 $\times$ 250 mm) column; Flow rate: 0.7 ml/min; Mobile phase composition: water:acetonitrile (ACN) gradient solvent system containing 0.1% formic acid: 0–30 min, 5–100% ACN; hold for 5 min, 100% ACN; 0.1 min, 100–5% ACN; hold for 1.9 min, 5% ACN; 6.1 min re-equilibration post-time, 5% ACN).

## Construction of scar-less gene deletion constructs

All of the genes in the pepteridine gene cluster were individually deleted using PCR to yield a complete series of clean deletion constructs designed to minimize polar effects on the operon and preserve necessary internal transcriptional and translational elements. PCR reactions (50 µl) were prepared using the pEplu2796 construct DNA as a template (~120 ng/µl). The primer pairs used for these reactions are listed in *Supplementary file 1B* (e.g., *Delplu2793F* and *Delplu2793R* - primer pair for the creation of a $\Delta plu2793$ construct). Primers were designed according to established protocols (*Liu and Naismith, 2008*). Briefly, each primer comprised two regions: one 3' region, which binds immediately after the genetic region to be deleted (bold-faced), and one 5' region, which binds immediately before the genetic region to be deleted. Standard molecular biology protocols as described above were employed. Deletion of the desired gene was validated by sequencing using the sequencing primer (*Supplementary file 1C*) immediately upstream of the deleted region ($\Delta plu2793$, pET28aUpstream; $\Delta plu2794$, Seq T7P*1; $\Delta plu2795$, Seq 2; $\Delta plu2796$, Seq 6; $\Delta plu2797$, Seq 11; $\Delta plu2798$, Seq 14; $\Delta plu2799$, Seq 15). Constructs possessing the desired deletion sequences were then fully sequence validated.

## Comparative metabolomic profiling and molecular networking

Comparative metabolomic analyses and pathway-targeted molecular networking were performed following established workflows with minimal modifications, detailed subsequently (*Figure 3*, *Figure 3—figure supplements 1* and *2*, *Table 1* and *Supplementary file 1A*) (*Vizcaino et al., 2014a*; *Vizcaino and Crawford, 2015*). Biosynthetic pathway expression was carried out in M9 minimal media supplemented with 5 g/l casamino acids, 0.5 g/l L-phenylalanine (Sigma-Aldrich, USA), and 25 µg/ml kanamycin. Expression cultures were inoculated at 1:1000 from stationary phase cultures grown overnight. Seven expression cultures were prepared. Six of these were biological replicates and one was a technical replicate that was solely used to monitor growth. As each sample set reached the desired $OD_{600}$ of 0.5–0.6, it was briefly placed on ice to cool. Once all sample sets had reached density and had sufficiently cooled, expression was then induced with 0.1 mM IPTG, and the culture sets were allowed to grow for 72 hr at 25°C and 250 rpm. The cell mass was then pelleted via centrifugation at 2000 $\times$ g for 20 min at 4°C. The supernatant from each sample was collected (~5 ml) and subsequently extracted with 6 ml of water-saturated butanol. The organic layer was collected and dried under reduced pressure on a Genevac HT-4X evaporation system. Dried extracts were stored under nitrogen at −80°C until use. Samples were prepared for HR-ESI-QTOF-MS analysis by resuspension in 200 µl methanol (LC-MS grade, Fisher, USA). Insoluble debris was removed by centrifugation at 20,000 $\times$ g for 5 min and 50 µl of the supernatant was placed in an HPLC vial for analysis. The remaining sample was dried and stored as previously described for future analysis. The Agilent iFunnel 6550 QTOF system was used for sample analysis. 2 µl of sample was injected and analyzed at 25°C and 0.7 ml/min on a Phenomenex Kinetex $C_{18}$ (100 Å) 5 µm (4.6 $\times$ 250 mm) column with a water:ACN gradient solvent system containing 0.1% formic acid: 0–30 min, 5–100% ACN; hold for 5 min, 100% ACN; 0.1 min, 100–5% ACN; hold for 1.9 min, 5% ACN; 6.1 min re-equilibration post-time, 5% ACN. Mass spectra were acquired in the range of 25–1700 *m/z* at a

scan rate of 1 spectra/s using Dual Agilent Jet Stream (AJS) ESI in positive mode. Source parameters were set as follows: drying gas temp, 225°C; drying gas flow, 12 l/min; nebulizer pressure, 35 psig; sheath gas temperature, 275°C; sheath gas flow, 12 l/min; fragmentor voltage, 125 V; skimmer voltage, 65 V; OCT 1 RF Vpp, 750 V; capillary voltage, 3500 V; nozzle voltage, 1000 V. Data were acquired and analyzed using MassHunter Workstation Data Acquisition (Version B.05.01, Build 5.01.5125.1, Agilent Technologies) and MassHunter Qualitative Analysis (Version B.06.00, Agilent), respectively. Comparison of the six biological replicates for each sample set allowed for selection of the five with the highest overall chromatographic similarity for further processing. Molecular feature extraction was performed with the following parameter alterations: peak spacing tolerance, 0.0025 $m/z$ plus 7 ppm; limit assigned charge states to a maximum of, 1; restrict charge states to, 1. In importing data into Mass Profiler Professional (MPP, Agilent), 'minimum number of ions' was set to 1. In performing the initial abundance analysis on the samples, a minimal normalized abundance of 18 was set. For controls, a molecular feature found in any one of the five samples was tabulated. Conservatively for wild-type, all five biological replicates had to possess the molecular feature for it to be tabulated. To ensure that all molecular features present in mutant strains throughout the five biological replicates were tabulated, a feature was tabulated if found in any one of the five biological replicates. Molecular features found in control samples were removed from both the wild-type and individual mutant feature lists. Wild-type pathway molecular features were subsequently compared withmutant pathway molecular features. Features present in the wild-type and absent in a given mutant were deemed to be dependent on that given mutant. An inclusion list was generated around these features for subsequent pathway-targeted tandem MS (MS$^2$) analysis. MS$^2$ analysis was run following the previously described acquisition method with some changes: nebulizer pressure, 50 psig; fragmentor voltage, 200 V. Auto MS$^2$ data collection was used with spectra being acquired at 1 spectra/s in the mass range of 25–1700 $m/z$. Fixed collision energies were set at 0, 25, 40, 45, 50, and 100 V. Auto MS$^2$ precursors were limited to those determined to be fully pathway-dependent or dependent on a given enzyme in the pathway. A maximum of 20 precursor ions were analyzed per cycle and a minimum precursor threshold was set at 10,000 counts (absolute) or 0.01% (relative). Isotope models were inactivated and scan speed was varied based on precursor abundance with a target of 25,000 counts/spectrum. The MS$^2$ accumulation time limit was employed. All target masses were allowed within a 10 ppm range and a 0.5 min retention time margin of error. Additionally, Auto MS$^2$ was used without a preferred ion list on the wild-type sample to establish an untargeted molecular network. The collision energy dependence of the fragmentation patterns of select molecular features was assessed, and we determined that 40 V was optimal for fragmentation of the pepteridines. After data conversion to the mzXML file format, each file was edited to remove non-optimal collision energies while maintaining the preferred 40 V collision energy. This was done as the presence of multiple fragmentation patterns for a single molecular feature can result in the presence of said feature multiple times in the molecular network. In establishing molecular networks through the Global Natural Products Social Molecular Networking Platform (GnPS, http://gnps.ucsd.edu), the following parameters were varied from their default values: parent mass tolerance, 0.001 Da; ion tolerance, 0.5 Da; min cos, 0.5; all filtering was disabled and filter below STD DEV and min peak intensity were both set to 0. Cytoscape (Cytoscape Consortium, USA) was used to visualize and edit the final networks. Parent ion masses networked in GnPS that did not match those present on our initial inclusion list based on precise retention time and high-resolution mass data, were removed from the network. Additionally, all features present in the wild-type inclusion list were validated against the control. Any features that were discovered to be false positives were subsequently removed from the network. When single nodes lacking connectivity were generated during this process, these nodes were removed from the network. The charge state for each feature was evaluated manually against the computationally generated charge state and any inaccuracies were corrected. For all networks, the thickness of the edges connecting nodes is representative of the strength of interaction between the nodes, with thicker lines denoting stronger interactions. The minimum thickness was set at 2 pts representing a cosine score of 0.5 while the thickest possible edge possessing a cosine score of 1 was set to 15 pts.

## Validation of pepteridine biosynthetic dependence on NRPS carrier protein machinery

To decouple the NRPS machinery from that of the fused dehydrogenase E2 subunit in Plu2796, a point mutation in the wild-type pathway was generated (pEplu2796-S434A). In NRPS logic, the biosynthetic pathway intermediates are ferried through the various rounds of enzymatic modification via attachment to a phosphopantetheine arm. This phosphopantetheine tether is post-translationally coupled to a conserved Ser residue. Removal of this Ser destroys carrier protein activity. Bioinformatic assessment of Plu2796 demonstrated a phosphopantetheinyl binding site in the T domain. This conserved Ser was identified as S434 by pattern matching with known amino acid motifs indicative of a phosphopantetheinyl attachment site. Using the mutagenesis strategy outlined in 'Construction of scar-less deletion constructs' with the modifications detailed subsequently, S434 was mutated to A434. This mutation prevents the attachment of phosphopantetheine thereby inactivating the NRPS functionality of the Plu2796 enzyme. The primers *PluS434AF* and *PluS434AR* were used in the PCR reactions. The wild-type pEplu2796 vector was used as a template. Successful point mutagenesis was assessed via sequencing using the *Seq 8* primer (**Supplementary file 1C**). The pEplu2796-S434A construct was then fully sequence validated over the region harboring the biosynthetic pathway. This construct was transformed into *E. coli* BAP1 (see 'Preparation of constructs for heterologous expression studies'). Subsequently, LC-HR-ESI-QTOF-MS analysis was performed on this construct in comparison with the wild-type (pEplu2796), negative control (pET28a), and Δ2796 constructs according to the instrument parameters and chromatography method outlined in 'Comparative metabolomic profiling and molecular networking.' Comparison of extracted ion chromatograms demonstrated the dependence of pepteridine A and B on the NRPS thiolation domain present in Plu2796. Extracted ion chromatograms of the S434A mutant match with those for the full deletion of Plu2796. All experiments were conducted in triplicate.

## Larger-scale growth and organic extraction

A 5 ml LB liquid culture supplemented with 25 µg/ml kanamycin was initiated by inoculation of a single colony of the *E. coli* BAP1 strain carrying the wild-type pepteridine construct, pEplu2796. After overnight growth at 37°C and 250 rpm, the culture was used to seed additional 32 × 5 ml LB cultures, which were further incubated at 37°C and 250 rpm for 18 hr. Each of the 32, 5 ml cultures was used to inoculate one of 32, 1 l cultures containing M9 minimal medium supplemented with 25 µg/ml kanamycin. These cultures were incubated at 37°C and 250 rpm until the $OD_{600}$ reached 0.5–0.6. Pathway expression was initiated via the addition of 0.1 mM IPTG, and the cultures were further incubated at 25°C and 250 rpm for 72 hr. The whole 32 l culture volume was centrifuged at 14,000 *g* and 4°C for 30 min, and the supernatant was lyophilized (~7 days). The dried sample was extracted with a total of 2 l of 50% aqueous methanol, filtered, and evaporated under reduced pressure to yield the crude material (approximately 4.0 g).

## Isolation and purification of compounds

The crude organic extract (4.0 g) was subjected to flash $C_{18}$ column chromatography (300 g) with a step gradient elution (0%, 10%, 20%, 40%, 60%, and 100% methanol in water). The 20% methanol fraction was further fractionated using an Agilent Prepstar HPLC system (Agilent Polaris $C_{18}$-A 5 µm (21.2 × 250 mm) column) with a linear gradient elution (5–50% methanol in water over 60 min, 8 ml/min, 1 min fraction collection window). A combined fraction (27 + 28) was subsequently isolated by reversed-phase HPLC (Phenomenex Luna $C_{18}$ $C_8$(2) (100 Å) 10 µm (10.0 × 250 mm) column) with a linear gradient elution (5–100% ACN in water over 60 min, 4 ml/min) to yield impure compound **1**. Compound **1** ($t_R$ = 19.85 min, 0.8 mg) was then purified by reversed phase $C_8$-HPLC (Phenomenex Luna $C_8$(2) (100 Å) 10 µm (10.0 × 250 mm) column) with isocratic separation (10% methanol in water, 2 ml/min), followed by final purification over an Agilent Phenyl-Hexyl 5 µm (9.4 × 250 mm) column using isocratic separation (5% ACN in water, 2 ml/min) to yield 0.8 mg of pure **1**. For compound **2**, the 10% methanol fraction from the flash $C_{18}$ column chromatography step was separated on a semi-preparative reversed-phase HPLC system (Agilent Polaris $C_{18}$-A 5 µm (21.2 × 250 mm) column) using isocratic separation (5% ACN in water with 0.01% TFA, 8 ml/min, 1 min fraction collection window). A pooled fraction (13 + 14) was then isolated by reversed-phase HPLC (Phenomenex Luna $C_{18}$(2) (100 Å) 5 µm (4.6 × 150 mm) column) using an isocratic solvent system (5% ACN in water over

30 min, 2 ml/min) to afford impure compound **2** ($t_R$ = 15.86 min). Final purification of compound **2** (1.2 mg, $t_R$ = 6.51 min) was carried out over an Agilent Phenyl-Hexyl 5 μm (9.4 × 250 mm) column using isocratic separation (5% ACN in water, 2 ml/min). Additionally, a combined fraction (10 + 11) containing compound **3** was further isolated by reversed-phase HPLC (Phenomenex Luna C$_8$(2) (100 Å) 10 μm (10.0 × 250 mm) column) using an isocratic solvent system (10% ACN in water over 30 min, 2 ml/min). Compound **3** ($t_R$ = 9.5 min) was then purified via an Agilent Phenyl-Hexyl 5 μm (9.4 × 250 mm) column eluting with an isocratic solvent system (3% methanol in water, 2 ml/min) to yield 3 mg of pure material.

Pepteridine A (**1**): white powder; UV (CH$_3$OH) $\lambda_{max}$ (log $\varepsilon$) 282 (3.68), 222 (3.85) nm; $^1$H and $^{13}$C NMR spectra, see **Table 2**; HR-ESI-QTOF-MS [M+H]$^+$ $m/z$ 224.1141 (calcd for C$_9$H$_{14}$N$_5$O$_2$, 224.1147).

Pepteridine B (**2**): white powder; UV (CH$_3$OH) $\lambda_{max}$ (log $\varepsilon$) 282 (3.48), 222 (3.75) nm; $^1$H and $^{13}$C NMR spectra, see **Table 2**; HR-ESI-QTOF-MS [M+H]$^+$ $m/z$ 210.0985 (calcd for C$_8$H$_{12}$N$_5$O$_2$, 210.0991).

7,8-Dihydroxanthopterin (**3**): white powder; UV (CH$_3$OH) $\lambda_{max}$ (log $\varepsilon$) 310 (2.99), 272, (3.25), 222 (3.54) nm; $^1$H NMR (DMSO-$d_6$, 600 MHz) $\delta$ 10.32 (1H, br s, 3-NH), 8.80 (1H, br s, 5-NH), 6.60 (1H, br s, 8-NH), 6.14 (2H, br s, 2-NH$_2$), 3.73 (2H, s, H-7); $^{13}$C NMR (DMSO-$d_6$, 100 MHz) $\delta$ 161.9 (C-6), 154.4 (C-4), 152.0 (C-2), 150.7 (C-8a), 93.4 (C-4a), 46.0 (C-7); HR-ESI-QTOF-MS [M+H]$^+$ $m/z$ 182.0631 (calcd for C$_6$H$_8$N$_5$O$_2$, 182.0678).

Pterin (**4**): pale yellow powder; UV (CH$_3$OH) $\lambda_{max}$ (log $\varepsilon$) 330 (2.48), 272 (3.26), 226 (3.46) nm; HR-ESI-QTOF-MS [M+H]$^+$ $m/z$ 164.0574 (calcd for C$_6$H$_6$N$_5$O, 164.0572).

## Structural characterization of metabolites

The chemical structures of metabolites **1–4** were identified by analyses of NMR and HR-ESI-QTOF-MS data; and spectral comparison with validated standards. The structure of compound **3**, which was elucidated by 1D- and 2D- (gCOSY and gHMBCAD) NMR experiments, was identified as the 7,8-dihydroxanthopterin (**3**) and supported by comparing its NMR spectral data with a commercial standard of **3**. The structure of pterin (**4**) was unambiguously characterized by HR-ESI-QTOF-MS data, HPLC co-injection with standard pterin, and by comparison of the UV absorption spectral data with that of a pterin commercial standard. Compound **1** was isolated as a white powder. The molecular formula was assigned as C$_9$H$_{13}$N$_5$O$_2$ ([M+H]$^+$ at $m/z$ 224.1141) based on HR-ESI-QTOF-MS spectroscopic data. The $^1$H NMR spectral data recorded in DMSO-$d_6$ displayed two NH protons [$\delta_H$ 10.07 (1H, br s), 6.96 (1H, br s)], an NH$_2$ proton [$\delta_H$ 6.25 (2H, br s)], three methylene groups [($\delta_H$ 4.54, 2.33), ($\delta_H$ 3.30, 2.79), ($\delta_H$ 2.57, 2.15)], and one methyl triplet ($\delta_H$ 0.88). Interpretation of the HSQC data coupled with $^{13}$C NMR spectral data showed a total of nine signals, which allowed us to assign all protons to the four directly bonded carbons ($\delta_C$ 41.9, 38.5, 26.5, and 9.7), together with the resonances of five quaternary carbons ($\delta_C$ 174.0, 157.3, 154.6, 153.1 and 93.1), suggesting the presence of a 2-amino-4-oxo-tetrahydropteridine moiety and a propionyl group. The sequential COSY correlations from a NH proton ($\delta_H$ 6.96) to a methylene group ($\delta_H$ 4.54, 2.33) established a dimethylene diamine-type partial structure and the COSY cross-peaks between a triplet methyl signal ($\delta_H$ 0.88) and a methylene group ($\delta_H$ 2.57, 2.15) also supported the presence of the propionyl group. The HMBC correlations from a NH proton ($\delta_H$ 6.96) and a methylene group ($\delta_H$ 2.57, 2.15) to a quaternary carbon ($\delta_C$ 93.1) allowed us to construct a 1,2,3,4-tetrahydropyrazine ring system and the three bond HMBC correlation sharing a carbonyl amide ($\delta_C$ 174.0) from a methyl signal ($\delta_H$ 0.88) and a methylene group ($\delta_H$ 4.54, 2.33) led to attachment of the propionyl group to the tetrahydropyrazine ring via N-acylation at the 5-position of the pteridine ring. NOESY interpretation and the presence of the shifted methylene protons ($\delta_H$ 4.54, 2.33), which are adjacent to the amide group, supported formation of the N-acyl group corresponding to the *cis*-amide conformation in the rotamer system. The full structure of compound **1** was unambiguously characterized by the HR-ESI-QTOF-MS data analysis. Compound **2** was also isolated as a white powder. HR-ESI-QTOF-MS analysis showed that the molecular formula of **2** was C$_8$H$_{11}$N$_5$O$_2$ ([M+H]$^+$ $m/z$ 210.0985) possessing a 14 Da mass difference, which could arise from loss of CH$_2$ from **1**. The 1D- ($^1$H and $^{13}$C) NMR spectral data of **2** were almost identical to that of **1** except for the absence of the CH$_2$ signal, indicating the presence of an acetyl group instead of a propionyl group characterized from the NMR interpretation of compound **1**, which is supported by HMBC correlation from a singlet methyl proton ($\delta_H$ 1.97) and a methylene group ($\delta_H$ 4.52, 2.32) to a carbonyl amide ($\delta_C$ 170.6).

## Precursor feeding experiments

α-Ketobutyrate and pyruvate were purchased from Sigma-Aldrich. *E. coli* BAP1 transformed with the pepteridine pathway (pEplu2796) was plated onto LB agar containing 25 µg/ml kanamycin and grown overnight at 37°C. A single, well-defined colony was selected and cultured in 5 ml LB medium supplemented with 25 µg/ml kanamycin overnight at 37°C and 250 rpm. Five milliliters of M9 minimal media supplemented with 25 µg/ml kanamycin and either filter sterilized α-ketobutyrate or pyruvate at multiple concentrations were inoculated with 25 µl of overnight culture (1:200) and incubated at 37°C until the $OD_{600}$ was between 0.5 and 0.6. The M9 cultures were then induced by addition of 0.1 mM IPTG and incubated at 25°C and 250 rpm for 72 hr. The cultures were centrifuged (2000 *g*, 20 min, 4°C), and the supernatants were extracted with water-saturated butanol (1 × 5 ml). The organic layers were dried under reduced pressure using a Genevac HT-4X evaporation system, and the crude materials were resuspended in 100 µl of 50% aqueous methanol. The resuspensions were then analyzed by single quadrapole LC/MS (Column: Phenomenex Kinetex $C_{18}$ (100 Å) 5 µm (4.6 × 250 mm) column; flow rate: 0.7 ml/min; mobile phase composition: water:ACN gradient solvent system containing 0.1% formic acid: 0–30 min, 5–100% ACN; hold for 5 min, 100% ACN; 0.1 min, 100–5% ACN; hold for 1.9 min, 5% ACN; 6.1 min re-equilibration post-time, 5% ACN). The feeding experiments were conducted in triplicate.

## Isotope labeling experiments

$^{13}C_4$-α-ketobutyrate was purchased from Cambridge Isotope Laboratories (USA). A single colony of *E. coli* BAP1 carrying the pepteridine pathway (pEplu2796) was inoculated into 5 ml LB medium supplemented with 25 µg/ml kanamycin. Three biological replicates were prepared, and the cultures were incubated at 37°C and 250 rpm overnight. Five milliliters of M9 minimal media supplemented with 25 µg/ml kanamycin and filter sterilized $^{13}C_4$-α-ketobutyrate (1.2 g/l) was inoculated with 25 µl of overnight culture and incubated at 37°C until the $OD_{600}$ was between 0.5 and 0.6. The M9 cultures were then induced by addition of 0.1 mM IPTG and incubated at 25°C and 250 rpm for 72 hr. The cultures were centrifuged (2000 × *g*, 20 min, 4°C), and the supernatants were extracted with water-saturated butanol (1 × 5 ml). The organic layers were dried under reduced pressure using a Genevac HT-4X evaporation system, and the crude materials were resuspended in 100 µl of 50% aqueous methanol. The resuspensions were then analyzed on the Agilent iFunnel 6650 QTOF system (Column: Phenomenex Kinetex $C_{18}$ (100 Å) 5 µm (4.6 × 250 mm) column; flow rate: 0.7 ml/min; mobile phase composition: water:ACN gradient solvent system containing 0.1% formic acid: 0–30 min, 5–100% ACN; hold for 5 min, 100% ACN; 0.1 min, 100–5% ACN; hold for 1.9 min, 5% ACN; 6.1 min re-equilibration post-time, 5% ACN).

## Pepteridine metabolites analysis in M-form and P-form phenotypic variants

Genetically locked *P. luminescens* in the M- and P- forms (*Somvanshi et al., 2012*) were grown on Luria-Bertani agar plates at 30°C. Single colonies of M- and P-forms were inoculated into individual 5 ml LB liquid medium and cultivated in a shaking incubator for 48 hr (30°C, 250 rpm). Five milliliters of each culture broth were then transferred to 1 l scale of M9 minimal medium (0.2% (w/v) glucose, 2 mM $MgSO_4$, and 0.1 mM $CaCl_2$) supplemented with 5 g/l casamino acids and cultivated at 30°C and 250 rpm. After 72 hr, 1 l each of M- and P-form culture broths were centrifuged at 14,000 × *g* for 20 min, and the supernatant was then extracted with butanol (2 × 1 l). The butanol-soluble fractions were separately dried under reduced pressure. The samples were subsequently resuspended in 5 ml methanol and 1–2 µl of each sample was injected for HR-ESI-QTOF-MS analysis (Column: Phenomenex Kinetex $C_{18}$ (100 Å) 5 µm (4.6 × 250 mm) column; flow rate: 0.7 ml/min; mobile phase composition: water:acetonitrile (ACN) gradient solvent system containing 0.1% formic acid: 0–30 min, 5–100% ACN; hold for 5 min, 100% ACN; 0.1 min, 100–5% ACN; hold for 1.9 min, 5% ACN; 6.1 min re-equilibration post-time, 5% ACN). Extracted ion count chromatograms were obtained by extracting with *m/z* 224.1147 and 210.0991 corresponding to pepteridines A and B, respectively, with a 10 ppm mass window. The butanol-soluble fraction sample from genetically locked P-form was used in co-injection experiments with standards for confirmation by HR-ESI-QTOF-MS using the same analytical methods.

## Deletion of the pepteridine genetic locus in *P. luminescens*

Allelic-exchange mutagenesis was used to excise the pepteridine genetic locus in *P. luminescens* TT01 (*pluCDS3313775R-plu2799*)(*Crawford et al., 2010*; *Kontnik et al., 2010*). Approximately 1.5 kB of both upstream and downstream genomic sequences were amplified using the primer pairs *Up-F* and *Up-R*; and *Dwn-F* and *Dwn-R*, respectively. PCR reactions (50 µl) were prepared using Q5 High-Fidelity DNA Polymerase according to the manufacturer's protocols with the inclusion of 5% (v/v) DMSO and 1 µl of confluent, wild-type *P. luminescens* TT01 as template. Products were purified using the QIAquick PCR Purification Kit following the manufacturer's protocol. Overlap extension PCR was then used to fuse the fragments (*Ho et al., 1989*). The homologous recombination cassette and pDS132 were both digested with SacI-HF (NEB) (*Philippe et al., 2004*). rSAP (NEB) was included in the pDS132 digest to dephosphorylate the vector. The linearized vector and cassette were ligated using T4 DNA ligase. Ligation products were transformed into chemically competent *E. coli* DH5α λpir using a standard heat-shock transformation procedure. Positive transformats were selected by plating 150 µl of the outgrowth onto LB agar plates supplemented with 25 µg/ml chloramphenicol (American Bioanalytical) and grown overnight at 37°C. Colony PCR (cPCR) was used to screen for positive constructs. cPCR reactions were carried out as described above using the primer pair *pDS-F* and *pDS-R*. A positive construct was subsequently sequence validated using primers *pDS-F*, *pDS-R*, *CDSSeq-1*, *CDSSeq-2*, *99Seq-1*, and *99Seq-2*. This construct was transformed by heat-shock into the diaminopimelic acid (DAP) auxotrophic donor strain *E. coli* WM6026 λpir (*Blodgett et al., 2007*) as described above with additional inclusion of 0.3 mM DAP in the LB agar plates. This strain along with wild-type *P. luminescens* TT01 and *P. luminescens* Rif$^r$Δ*hexA* (generously provided by Professor David Clarke) were grown to confluency in LB media at 37°C and 30°C, respectively, and 250 rpm. All overnight growths were subcultured 1:1000 into LB and grown to $OD_{600} \approx 0.6$; mixed at 1:1 and 1:4 (donor:recipient) ratios; and filtered through a 0.4 µm sterile filter. Filter mating was carried out overnight at 30°C on LB agar supplemented with 0.3 mM DAP. The outgrowth was then resuspended in LB and streaked onto LB agar containing chloramphenicol (25 µg/ml). Colored colonies indicative of *Photorhabdus* were selected and streaked onto LB agar supplemented with 5% sucrose for SacB counterselection. Positive colonies were re-streaked three times and single colonies were validated by cPCR. Positive cPCR products were purified using the QIAquick PCR Purification Kit and sequence validated using the primers *LocUp*, *LocDown*, *CDSSeq-1*, *CDSSeq-2*, *99Seq-1*, and *99Seq-2*.

## Proteomics

*P. luminescens* wild-type and Δ*hexA* strains encoding or lacking the pepteridine synthesis pathway were grown in 5 ml hemolymph-mimetic medium (5 g yeast extract, 10 g NaCl, and proteinogenic amino acids based on hemolymph concentrations) for twenty-four hr at 30°C and 250 rpm. 200 µl pellets of strains were prepared in biological triplicate and frozen at −80°C. Upon thawing, sample processing (protein extraction, alkylation, and trypsin digest) was performed as in *Lajoie et al. (2013)*. Peptides were desalted using a $C_{18}$ MacroSpin column (The Nest Group). Samples were then dried in a centrifugal vacuum concentrator and dissolved in 22 µl 70% formic acid:0.1% tri-fluoroacetic acid mixed 3:8 volumetrically. In the same buffer, samples were further diluted to 0.5 µg/µl based on $A_{280}$ measurements, and 2.5 µg of peptides (5 µl) were analyzed using an ACQUITY UPLC M-Class (Waters) paired with a Q Exactive Plus (Thermo) mass spectrometer. Column parameters, gradient profiles, and settings for mass spectrometry are described in *Ferdaus et al. (2016)*.

## Proteomic data searches and analysis

The *P. luminescens* annotated proteome was downloaded from Uniprot (proteome ID UP000002514) and used for mass spectra searches using MaxQuant v.1.5.1.2 (Cox & Mann) with Cys carbamidomethyl fixed modification and Asn/Gln deamidation, Met oxidation, Ser/Thr/Tyr phosphorylation, and N-terminal acetylation variable modifications. Search parameters considered peptides resulting from two or fewer missed tryptic cleavages and peptides at least five residues long with a 1% false discovery rate. Using Perseus software v.1.5.0.15 (*Tyanova et al., 2016*), averaged label-free quantification intensities for sample replicates were compared between two samples by two-tailed T test, with significance cutoffs for protein intensity differences established by either

permutation-based false discovery rate (FDR = 0.01 with a 2-fold minimum intensity difference) or p<0.05, both reported in *Figure 8—source data 1–6*.

## General

UV/Vis spectra were obtained on an Agilent (Agilent Technologies, USA) Cary 300 UV-visible spectrophotometer with a path length of 10 mm. [1]H and 2D- (gCOSY, gHSQCAD, gHMBCAD and NOESY) NMR spectral data were recorded on an Agilent 600 MHz NMR spectrometer equipped with a cold probe. [13]C NMR spectral data were recorded at 100 MHz on an Agilent NMR spectrometer. Flash column chromatography was carried out on Lichroprep RP-18 (40–63 µm, Merck, USA). Routine HPLC analysis was performed on an Agilent 1260 Infinity system with a Phenomenex (USA) Luna $C_{18}$(2) (100 Å) 5 µm (4.6 × 150 mm) column and a Photo Diode Array (PDA) detector. The separation and purification of metabolites were performed using an Agilent Prepstar HPLC system with an Agilent Polaris $C_{18}$-A 5 µm (21.2 × 250 mm) column, a Phenomenex Luna $C_{18}$(2) or $C_8$(2) (100 Å) 10 µm (10.0 × 250 mm) column, and an Agilent Phenyl-Hexyl 5 µm (9.4 × 250 mm) column. Low-resolution electrospray ionization mass spectrometry (ESI-MS) data were measured on an Agilent 6120 Quadrupole LC/MS system with a Phenomenex Kinetex $C_{18}$ (100 Å) 5 µm (4.6 × 250 mm) column. High-resolution ESI-MS data were obtained using an Agilent iFunnel 6550 QTOF (quadrupole time-of-flight) MS instrument fitted with an electrospray ionization (ESI) source coupled to an Agilent 1290 Infinity HPLC system.

## Acknowledgements

Our work on the discovery of bioactive metabolites in host-bacteria interactions was supported by the National Institutes of Health (National Cancer Institute grant 1DP2-CA186575 and National Institute of General Medical Sciences grant R00-GM097096), the Searle Scholars Program (grant 13-SSP-210), and the Damon Runyon Cancer Research Foundation (grant DRR 39–16). CEP was supported in part by a National Institutes of Health Chemistry Biology Interface Training Grant (5T32GM067543-12). JR was supported by the National Institute of General Medical Sciences grant R01GM117230 and KWB was supported by National Science Foundation grant DGE-1122492. We thank Yue-Wei Ge for preliminary analysis of pterin and 7,8-dihydroxanthopterin.

## Additional information

### Funding

| Funder | Grant reference number | Author |
| --- | --- | --- |
| National Institutes of Health | Chemistry Biology Interface Training, 5T32GM067543-12 | Corey E Perez |
| National Science Foundation | DGE-1122492 | Karl W Barber |
| National Institute of General Medical Sciences | R01GM117230 | Jesse Rinehart |
| The Searle Scholars | 13-SSP-210 | Jason M Crawford |
| Damon Runyon Cancer Research Foundation | DRR 39-16 | Jason M Crawford |
| National Institute of General Medical Sciences | R00-GM097096 | Jason M Crawford |
| National Cancer Institute | 1DP2-CA186575 | Jason M Crawford |

The funders had no role in study design, data collection and interpretation, or the decision to submit the work for publication.

### Author contributions

HBP, CEP, Conceptualization, Validation, Investigation, Methodology, Writing—original draft, Writing—review and editing; KWB, Investigation, Methodology, Writing—original draft, Writing—review

and editing; JR, Funding acquisition, Investigation, Writing—review and editing; JMC, Conceptualization, Supervision, Funding acquisition, Writing—review and editing

## Author ORCIDs

Hyun Bong Park, http://orcid.org/0000-0002-2986-3833
Jason M Crawford, http://orcid.org/0000-0002-7583-1242

## Additional files

### Supplementary files

• Supplementary file 1. Molecular feature list and primers used. (A) Molecular feature list dependent on the presence of the wild-type pathway. (B-D) Primers used.

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
