## [Decision Letter]

[Editors’ note: a previous version of this study was rejected after peer review, but the authors submitted for reconsideration. The first decision letter after peer review is shown below.]

Thank you for submitting your work entitled "Genome mining unearths a hybrid nonribosomal peptide synthetase-like-pteridine synthase biosynthetic gene cluster" for consideration by *eLife*. Your article has been favorably evaluated by Richard Losick as the Senior Editor and three reviewers, one of whom, Jon Clardy (Reviewer #1), is a member of our Board of Reviewing Editors.

Our decision has been reached after consultation between the reviewers. Based on these discussions and the individual reviews below, we regret to inform you that your work will not be considered further for publication in *eLife*.

All three reviewers were enthusiastic about many aspects of your manuscript, especially the genome mining effort that led to identifying the pepteridine cluster along with the rigor of the chemical and biochemical characterization of the pepteridines and their pathway. Those features provided meticulously characterized and important chemical and biochemical insights.

However, the referees also perceived some shortcomings in the manuscript. One was the lack of any significant biological activity. This lack could be related to your not screening in physiologically or ecologically relevant assays. It could also be related to a second issue: Are the pepteridines the intended product of the pathway, or could they be the partial products produced by an incomplete pathway. The referees feel that you 1) need to do additional experiments to establish, or try to establish, pepteridine production in the native strain, and 2) address the shortcomings noted in the biological activity section. You could either find a significant level of activity in a therapeutically relevant assay, or activity in an ecologically relevant assay, or deleting the entire section. Finally, the suggestion of additional bioinformatic analysis to establish the distribution of the cluster would add to the manuscript's significance.

If you elect to revise the manuscript to address these issues, we would be pleased to look at a new submission.

*Reviewer #1:*

The manuscript by Park et al. describes a genome mining analysis that revealed an unconventional biosynthetic pathway that incorporates features from two well-known biosynthetic pathways – nonribosomal (NRPS) peptide synthetase and pteridine synthase – to create a hybrid pathway that makes intriguing small molecules called "pepteridines." Given the intense research activity in both genome mining for small molecule pathways and NRPS pathways, this discovery will be important to many researchers.

NRPS pathways make many important therapeutic agents and pteridines are important redox active cofactors, this hybrid pathway, or its relatives, could produce molecules with important biological and/or medical activity. The discovery of the pepteridines was made possible by a genome mining approach based on genome synteny rather than the more conventional similarity to previously known pathway approach. The pathway was analyzed using heterologous expression, gene deletion, mutagenesis, and 'pathway-targeted molecular networking.' The pepteridines were thoroughly characterized spectroscopic, especially NMR, methods and chemical synthesis. All in all this is an impressive report with important lessons on research carried out at the highest technical level. There are, of course, some issues that need to be addressed.

The brief section on biological activity is well below the standard of the rest of the manuscript. The authors report "moderate cytotoxic activities against a colon cancer cell line (HCT-116, 77.57 and 79.26 mean growth percent for (1) and (2), respectively, whereas 2 selectively exhibited moderate activity against a breast cancer cell line (TD-47D, 77.11 mean growth percent." These data, which were cherry picked from a much larger NCI screen (Figure 4—figure supplements 14 and 15), show little selectivity and little potency, i.e. they would not be identified by the NCI screen as worthy of additional studies. The antifungal activity, ~200 µM MIC, is essentially nonexistent, and another screen against a panel of kinases showed nothing of interest.

The authors also report speculative activity based on structural comparisons with molecules with known activity, but if associating molecular structures with biological activity were that easy, drug discovery would be a very different enterprise. The pepteridines are similar to a xanthine oxidase inhibitor, and screening them as pteridine reductase inhibitors would be a good idea as they are redox inactive analogs of the endogenous substrates. But such a screen wasn't done. The manuscript also points out that they could be MR1 (a MHC class I-like molecules) based on structural similarity, but this screen was also not done.

I think that the manuscript would be stronger without this section as it undercuts the important aspects of the report.

*Reviewer #2:*

Nonribosomal peptides and pteridines are both large and biologically significant classes of metabolites that are typically produced through their own independent biosynthetic machinery. Through the genome mining of *P. luminescens*, the authors of this current study have identified the first example of a mixed NRPS-pteridine biosynthetic gene cluster. By way of the heterologous expression of this new hybrid cluster, the authors produced, isolated, and structurally characterized two novel cytotoxic pteridine metabolites, pepteridine A and B. Gene deletion studies, molecular networking analyses, stable-isotope feeding experiments and molecule characterization studies all appear to have been done very well and support the authors' general conclusions. A few items that the authors might think about are outlined below:

1) It would be nice if the authors could discuss in more detail whether they believe pepteridine A and B are the intended metabolites produced by this gene cluster.

A) A brief discussion (in the manuscript) of the genes flanking this bioinformatically identified island would be helpful to convince the reader that the set of genes studied by the authors constitute an entire gene cluster.

B) 1 and 2 are minor metabolites observed in heterologous expression experiments using *E. coli*:BAP1. In this system the pteridine intermediate is produced at a much higher level. Are short acyl chains added to the pepteridine intermediate at such a low level because they are only the best available substrates in *E. coli*? As the authors mention in their introduction an NRPS A domain "selects its cognate amino acid from the available substrate pool". Is it possible that the pepteridine NRPS (TC domain) system naturally uses different substrates than those available in *E. coli*? Could this explain the very low yield of 1 and 2 and the absence of any biological activity? It would be helpful if the authors could discuss their thoughts on these topics.

The authors present a tremendous amount of high quality work in this manuscript and therefore the following topics are likely beyond the scope of this manuscript. With that said, it would be nice to know the following. Can 1 or 2 be found in the culture broth of *P. luminescens*? Are any other pteridine metabolites seen in *P. luminescens* cultures?

2) The pteridine island is not placed in the context of what exists in other sequenced genomes. Applying a similar but now targeted sequence-based genome mining approach to find any similar pepteridine clusters present in other bacteria might improve the broad interest of the article.

3) While there is noteworthy significance in the identification of a novel type of hybrid NRPS-pteridine biosynthetic cluster, the lack of any discernable biological activity somewhat reduces the impact of this article. The authors might consider studying the biosynthesis and antibacterial effects of pepteridine under anaerobic conditions. The effects of oxygen in the regulation of the biosynthetic enzymes and yield of on/off-pathway products in the pteridine molecule family are well document (i.e. molydopterin, cobalamin biosynthesis). More significant to the authors, given the close structure of pepteridine to other essential pteridines, they could function as inhibitors of essential bacterial pteridine biosynthetic and pteridine-dependent enzymes, but only under anaerobic conditions. For example, the inhibition of molydopterin (molybdenum cofactor) biosynthesis in *E. coli* and Mtb is only toxic under anaerobic conditions due to the requirement of the pterin cofactor for activity of the anaerobic respiration enzyme, nitrogen reductase (See Kenichi Yokoyama, PNAS, 2015 & Peter Schultz, PNAS, 2015). As *P. luminescens* is found primarily in the low oxygen-containing gut of nematodes (Bernhard Schink, FEMS Micro, 1996), it would not be surprising to find some its secondary metabolites, especially pteridine-based, were similarly inhibiting enzymes regulated by and critical for anaerobic respiration.

*Reviewer #3:*

Crawford and coworkers report genes for a hybrid biosynthetic pathway involving parts of a nonribosomal peptide synthetase and a pteridine synthase. They report two compounds (pepteridine A + B) produced by *E. coli* expressing this gene cluster. Although the structures of pepteridines have not been reported before, they are not novel (only acylated derivatives of a known pteridine) and do not show any remarkable biological activities. The biosynthetic pathway is somewhat peculiar, yet the model is highly speculative and needs to be supported by biochemical experiments. Overall, this paper lacks novelty that would be appealing to the broad readership of *eLife* and would be better suited for a specialized journal.

1) By heterologous expression and gene deletion the authors proved the function of the gene cluster. However, the biosynthetic model is solely based on bioinformatic analyses and isotope labeling. This does not clear the bar of a journal like *eLife* with a very broad readership.

2) Care must be taken when expressing gene clusters in heterologous hosts. How can the authors rule out the possibility that *E. coli* produces artifacts? *Photorhabdus* might use different building blocks that are not present in *E. coli*. There is ample precedence for such a scenario.

3) Beyond its existence, the reader does not learn anything about the biological function of the metabolites. The assayed bioactivities are weak. Do pepteridines play any ecological role? Are pepteridine-like compounds found in *Photorhabdus* in a particular biological context?

[Editors’ note: what now follows is the decision letter after the authors submitted for further consideration.]

Congratulations, we are pleased to inform you that your article, "Genome mining unearths a hybrid nonribosomal peptide synthetase-like-pteridine synthase biosynthetic gene cluster", has been accepted for publication in *eLife*.

The authors responded very positively to all reviewer comments and suggestions in the earlier review. The change in the biological data was especially impressive.

*Reviewer #1:*

The revised (technically resubmitted) version of this manuscript is greatly improved and suitable for publication in *eLife*. In particular some off topic material on biological assays has been removed and replaced with new results with a relevant model: proteomic studies on a genetic locus mutant. Figures have been modified to make the discovery approach clearer. Other less important issues have been appropriately addressed.

*Reviewer #2:*

With their revised manuscript the authors have addressed the vast majority of the key concerns raised by the reviewers in a thoughtful and concise manner. At this point I have no additional concerns. I believe the current manuscript can be published as is.

*Reviewer #3:*

Crawford and coworkers detail the investigation of an unusual biosynthetic gene cluster, identified in *Photorhabdus luminescens* by inference and genomic synteny. The putative products of the BGC were identified by heterologous expression and comparative metabolomics (molecular networking) in *E. coli*, revealing a set of pteridine natural products, two of which are previously not described N-acetyl and N-propionyl analogs. Of interest is the proposal and validation of the biosynthetic pathway by genetics and isotopic incorporation studies revealing a new strategy fusing primary and secondary metabolism elements. The pepteridines are validated as being produced in *Photorhabdus* in a phenotypic growth subtype variants. Insights into the chemical ecological roles of the pepteridines was demonstrated by showing effects of the encoding locus on protein production associated with quorum sensing in *Photorhabdus*.

This manuscript has been through a round of review and revision and is quite polished at this point. The comments of the previous reviewers have been substantially responded to and the conclusions are well supported by the data in the current revision. What is most compelling about this manuscript in terms of the broad readership and impact scale of *eLife* is the gene-to-functional scope of the piece. Genome mining often results in the revelation of a new structures but rarely in a credible functional role in an organism or ecosystem. This work models a technical approach that will guide many interested in this rapidly expanding area of research.

---

## [Author Response]

[Editors’ note: the author responses to the first round of peer review follow.]

*[…] However, the referees also perceived some shortcomings in the manuscript. One was the lack of any significant biological activity. This lack could be related to your not screening in physiologically or ecologically relevant assays. It could also be related to a second issue: Are the pepteridines the intended product of the pathway, or could they be the partial products produced by an incomplete pathway. The referees feel that you 1) need to do additional experiments to establish, or try to establish, pepteridine production in the native strain, and 2) address the shortcomings noted in the biological activity section. You could either find a significant level of activity in a therapeutically relevant assay, or activity in an ecologically relevant assay, or deleting the entire section. Finally, the suggestion of additional bioinformatic analysis to establish the distribution of the cluster would add to the manuscript's significance.*

*If you elect to revise the manuscript to address these issues, we would be pleased to look at a new submission.*

Over the last 6-7 months, we have conducted additional experiments and have revised our manuscript in accordance with the reviewers’ comments, requests, and recommendations.

Specifically, the *eLife* editors/reviewers welcomed our resubmission if we could address the following points: 1) metabolite production in the native strain; 2) bolstering or deleting our prior in vitro bioactivity studies; and 3) providing bioinformatics support of the pathway’s phylogenetic distribution. We 1) established pepteridine production in the native strain, *P. luminescens*. Specifically, we identified pepteridine production only in one of the two established phenotypic variants of *P. luminescens* using genetically “locked” variant strains. These experiments connect pepteridine structure to bacterial phenotypic variation status. We 2) deleted the in vitro biological characterization section and added a new quantitative proteomic section to determine the genomic island’s signaling contributions in the producer strain. This focused our revised manuscript on cellular pathway analysis, the major and strong themes of the paper, and added two new authors. We 3) added additional bioinformatics statements in regards to the detection of the unprecedented hybrid enzyme Plu2796. In sum, we believe that we have suitably addressed the three perceived shortcomings of our original submission. Additionally, we believe the innovative review process at *eLife* has substantially strengthened our study.

*Reviewer #1:*

*[…] The brief section on biological activity is well below the standard of the rest of the manuscript. The authors report "moderate cytotoxic activities against a colon cancer cell line (HCT-116, 77.57 and 79.26 mean growth percent for (1) and (2), respectively, whereas 2 selectively exhibited moderate activity against a breast cancer cell line (TD-47D, 77.11 mean growth percent." These data, which were cherry picked from a much larger NCI screen (Figure 4—figure supplements 14 and 15), show little selectivity and little potency, i.e. they would not be identified by the NCI screen as worthy of additional studies. The antifungal activity, ~200 µM MIC, is essentially nonexistent, and another screen against a panel of kinases showed nothing of interest.*

*The authors also report speculative activity based on structural comparisons with molecules with known activity, but if associating molecular structures with biological activity were that easy, drug discovery would be a very different enterprise. The pepteridines are similar to a xanthine oxidase inhibitor, and screening them as pteridine reductase inhibitors would be a good idea as they are redox inactive analogs of the endogenous substrates. But such a screen wasn't done. The manuscript also points out that they could be MR1 (a MHC class I-like molecules) based on structural similarity, but this screen was also not done.*

*I think that the manuscript would be stronger without this section as it undercuts the important aspects of the report.*

The in vitro biological assay section describing moderate inhibitory activities for the new pepteridine metabolites was deleted in the revised manuscript. To initiate functional cellular studies in *P. luminescens*, we generated a pepteridine genetic locus mutant in a wild-type background and in a hexA mutant background. We included hexA, as very recent global transcriptomic analysis suggested that the pathway was upregulated in hexA strains (Bode and co-workers referenced herein). We analyzed these strains by quantitative proteomics. In a wild-type background, little effect was observed (indeed, many pathways are cryptic under standard cultivation conditions). However, in a hexA background, the pepteridine genetic locus significantly affected the proteome. We show that the pathway is under the control of HexA located elsewhere in the genome and that the pathway positively affects pyrone quorum sensing and select secondary metabolic pathway enzyme production. These studies support that the pepteridine genetic locus contributes to chemical signaling, which may explain the moderate in vitro bioactivities we previously noted.

*Reviewer #2:*

*[…] 1) It would be nice if the authors could discuss in more detail whether they believe pepteridine A and B are the intended metabolites produced by this gene cluster.*

In the revised manuscript, we show that the pepteridines are produced in the pathogenic P-form phenotypic variant of *P. luminescens*, further supporting our proposal.

*A) A brief discussion (in the manuscript) of the genes flanking this bioinformatically identified island would be helpful to convince the reader that the set of genes studied by the authors constitute an entire gene cluster.*

We emphasized the genome synteny approach in the revised manuscript and have a dedicated supporting Figure showing the genomic island and its synteny-defined boundaries.

*B) 1 and 2 are minor metabolites observed in heterologous expression experiments using E. coli:BAP1. In this system the pteridine intermediate is produced at a much higher level. Are short acyl chains added to the pepteridine intermediate at such a low level because they are only the best available substrates in E. coli?*

We (and the field) regularly see poor production or poor transformations (bottlenecks) in *E. coli* as a heterologous host. Supplementation of short chain α-ketoacids such as pyruvate and α-ketobutylate led to substantial upregulation of the products. We also supplemented a variety of other α-ketoacids, such as the ones derived from branched chain amino acid degradation and others, and we did not observe an effect with these substrates. These studies importantly link pepteridine structure to α-ketoacid substrates rather than simple acyl-CoA substrates. In the revised manuscript, detection of the two products in *Photorhabdus* now further supports this enzyme selectivity. We provide a response below to reviewer 3 in regards to protein biochemistry, which further supports our assignment as well. We are confident in the substrate preferences defined in the manuscript.

*As the authors mention in their introduction an NRPS A domain "selects its cognate amino acid from the available substrate pool". Is it possible that the pepteridine NRPS (TC domain) system naturally uses different substrates than those available in E. coli? Could this explain the very low yield of 1 and 2 and the absence of any biological activity? It would be helpful if the authors could discuss their thoughts on these topics.*

The yield wasn’t so bad for heterologous expression in *E. coli*. We observed much better production with amino acid supplements, but we explain that these supplements hindered isolation efforts. Again, detection in *Photorhabdus* supports a similar transformation as in *E. coli*.

*The authors present a tremendous amount of high quality work in this manuscript and therefore the following topics are likely beyond the scope of this manuscript. With that said, it would be nice to know the following. Can 1 or 2 be found in the culture broth of P. luminescens? Are any other pteridine metabolites seen in P. luminescens cultures?*

We appreciate the reviewer’s understanding of the amount of work in this study. We experimentally address this point in the revised manuscript. We could successfully detect the pepteridine metabolites in *P. luminescens* (Figure 7). *Photorhabdus luminescens* undergoes stochastic phenotypic variation between the P-form and M-form, a finding that we previously described in Science with Todd Ciche. The P-form is pathogenic to insects and the M-form is thought to participate in the colonization of its nematode host. Between the two forms, the metabolites were only detected in the P- form phenotypic variant (Figure 7). These studies link pepteridine structure to phenotypic variation status.

*2) The pteridine island is not placed in the context of what exists in other sequenced genomes. Applying a similar but now targeted sequence-based genome mining approach to find any similar pepteridine clusters present in other bacteria might improve the broad interest of the article.*

We provide a statement in the revised manuscript that describes the limited genetic distribution of the unprecedented hybrid enzyme fusion Plu2796 (the defining enzyme of the pathway) in current protein databases (BlastP). This pathway is highly novel (which is why we targeted it), and it is remarkable to see how NRPS biochemistry is coupled to pterin biochemistry. Now, the reader will better understand its novelty and genetic distribution.

*3) While there is noteworthy significance in the identification of a novel type of hybrid NRPS-pteridine biosynthetic cluster, the lack of any discernable biological activity somewhat reduces the impact of this article. The authors might consider studying the biosynthesis and antibacterial effects of pepteridine under anaerobic conditions. The effects of oxygen in the regulation of the biosynthetic enzymes and yield of on/off-pathway products in the pteridine molecule family are well document (i.e. molydopterin, cobalamin biosynthesis). More significant to the authors, given the close structure of pepteridine to other essential pteridines, they could function as inhibitors of essential bacterial pteridine biosynthetic and pteridine-dependent enzymes, but only under anaerobic conditions. For example, the inhibition of molydopterin (molybdenum cofactor) biosynthesis in E. coli and Mtb is only toxic under anaerobic conditions due to the requirement of the pterin cofactor for activity of the anaerobic respiration enzyme, nitrogen reductase (See Kenichi Yokoyama, PNAS, 2015 & Peter Schultz, PNAS, 2015). As P. luminescens is found primarily in the low oxygen-containing gut of nematodes (Bernhard Schink, FEMS Micro, 1996), it would not be surprising to find some its secondary metabolites, especially pteridine-based, were similarly inhibiting enzymes regulated by and critical for anaerobic respiration.*

Thanks for the excellent suggestions. We tried these recommended experiments. However, the pepteridines did not show substantial antimicrobial activities under anaerobic conditions. Nor did they show inhibitory activity against NO synthase, xanthine oxidase, or dihydrofolate reductase. As recommended by reviewer 1, we deleted this in vitro bioactivity section. Pterin signaling is a known phenomenon, so we considered whether the pepteridine genetic locus has an alternative signaling role. As noted above, we included a new quantitative proteomics section in the revised manuscript. These studies focus the current manuscript on cellular metabolic and proteomic analysis associated with the unprecedented gene cluster.

*Reviewer #3:*

*Crawford and coworkers report genes for a hybrid biosynthetic pathway involving parts of a nonribosomal peptide synthetase and a pteridine synthase. They report two compounds (pepteridine A + B) produced by E. coli expressing this gene cluster. Although the structures of pepteridines have not been reported before, they are not novel (only acylated derivatives of a known pteridine) and do not show any remarkable biological activities. The biosynthetic pathway is somewhat peculiar, yet the model is highly speculative and needs to be supported by biochemical experiments. Overall, this paper lacks novelty that would be appealing to the broad readership of eLife and would be better suited for a specialized journal.*

The pepteridines are new metabolites encoded by a remarkable biosynthetic system. Yes, N-acylation of the pterin scaffold has been reported in synthetic studies. Indeed, we adapted these synthetic approaches to make the new metabolites in our original submission. Those in vitro studies, which were removed in the revised manuscript, were appropriately referenced.

1) By heterologous expression and gene deletion the authors proved the function of the gene cluster. However, the biosynthetic model is solely based on bioinformatic analyses and isotope labeling. This does not clear the bar of a journal like eLife with a very broad readership.

We believe the work is suitable for *eLife*. Many biosynthesis papers describe metabolites that were already published (i.e., a detailed discovery paper and a biosynthesis paper typically report what is requested by this reviewer). In fact, collectively, the reviewers are requesting a 3^rd^ functional or mode of action dimension, which in certain cases is a fleshed out 3^rd^ paper. Our revised, long, and extensive manuscript, which addresses reviewer 1 and 2’s comments, is much stronger than the original submission.

Genome mining is a hot topic and we believe that the paper should be appealing to a wide audience.

We have also shown that the C-domain does in fact catalyze the transfer in isolated in vitro protein biochemical studies to reconstitute pepteridine production (the cis-amide is observed as the product). We have probed the pterin selectivity and the acyl-selectivity, and both of these studies support our model here. We are fleshing these details-oriented in vitro studies out for a specialty biochemistry journal over the next year. As noted by reviewer 2, these studies are beyond the scope of the current manuscript. We agree.

*2) Care must be taken when expressing gene clusters in heterologous hosts. How can the authors rule out the possibility that E. coli produces artifacts? Photorhabdus might use different building blocks that are not present in E. coli. There is ample precedence for such a scenario.*

See above in regards to detection of the metabolites in *Photorhabdus*.

*3) Beyond its existence, the reader does not learn anything about the biological function of the metabolites. The assayed bioactivities are weak. Do pepteridines play any ecological role? Are pepteridine-like compounds found in Photorhabdus in a particular biological context?*

See above in regards to new genetic and quantitative proteomic analyses in *Photorhabdus*.